# Caveolin-1 temporal modulation enhances antibody drug efficacy in heterogeneous gastric cancer

Patrícia M. R. Pereira[1,2 ✉], Komal Mandleywala[1], Sébastien Monette [3], Melissa Lumish[4], Kathryn M. Tully[1,5], Sandeep Surendra Panikar [2], Mike Cornejo[1], Audrey Mauguen[6], Ashwin Ragupathi[1], Nai C. Keltee[2], Marissa Mattar[7], Yelena Y. Janjigian[4] & Jason S. Lewis [1,5,8,9,10 ✉]

Resistance mechanisms and heterogeneity in HER2-positive gastric cancers (GC) limit Trastuzumab benefit in 32% of patients, and other targeted therapies have failed in clinical trials. Using patient samples, patient-derived xenografts (PDXs), partially humanized biological models, and HER2-targeted imaging technologies we demonstrate the role of caveolin-1 (CAV1) as a complementary biomarker in GC selection for Trastuzumab therapy. In retrospective analyses of samples from patients enrolled on Trastuzumab trials, the CAV1-high profile associates with low membrane HER2 density and low patient survival. We show a negative correlation between CAV1 tumoral protein levels – a major protein of cholesterol-rich membrane domains – and Trastuzumab-drug conjugate TDM1 tumor uptake. Finally, CAV1 depletion using knockdown or pharmacologic approaches (statins) increases antibody drug efficacy in tumors with incomplete HER2 membranous reactivity. In support of these findings, background statin use in patients associates with enhanced antibody efficacy. Together, this work provides preclinical justification and clinical evidence that require prospective investigation of antibody drugs combined with statins to delay drug resistance in tumors.

[1] Department of Radiology, Memorial Sloan Kettering Cancer Center, New York, NY 10065, USA. [2] Department of Radiology, Mallinckrodt Institute of Radiology, Washington University School of Medicine, St. Louis, MO 63110, USA. [3] Laboratory of Comparative Pathology, Memorial Sloan Kettering Cancer Center, Weill Cornell Medicine, and The Rockefeller University, New York, NY, USA. [4] Department of Medicine, Memorial Sloan Kettering Cancer Center, New York, NY, USA. [5] Department of Pharmacology, Weill Cornell Medical College, New York, NY, USA. [6] Department of Epidemiology and Biostatistics, Memorial Sloan Kettering Cancer Center, New York, NY, USA. [7] Antitumor Assessment Core Facility, Molecular Pharmacology Program, Memorial Sloan Kettering Cancer Center, New York, NY, USA. [8] Molecular Pharmacology Program, Memorial Sloan Kettering Cancer Center, New York, NY, USA. [9] Department of Radiology, Weill Cornell Medical College, New York, NY, USA. [10] Radiochemistry and Molecular Imaging Probes Core, Memorial Sloan Kettering Cancer Center, New York, NY, USA. ✉email: ribeiropereirap@wustl.edu; lewisj2@mskcc.org

Human epidermal growth factor receptor 2 (HER2) alterations, including overexpression, amplification, and other mutations, occur in breast and gastric cancer (GC)[1,2]. The anti-HER2 antibody Trastuzumab is the standard-of-care treatment for metastatic and early-stage HER2-positive breast cancer[2] and first-line therapy in combination with chemotherapy for GC[3]. Although several targeting agents are effective in treating HER2-positive breast tumors[1,2], not all tumors benefit from HER2-targeted therapies (reviewed in[2]) due to considerable differences in HER2 biology in different tumor types. Beyond Trastuzumab[3] and Trastuzumab deruxtecan[4], clinical trials have failed to demonstrate efficacy of other HER2-targeted therapies (Pertuzumab, TDM1) in the first and later treatment lines for GC[5,6]. The lack of remarkable achievements in GC suggests that the successes seen in breast cancer can not be replicated in several other tumor types, e.g., biliary tract, colorectal, non-small-cell lung and bladder cancers.

One of the reasons that not all GC respond to targeted therapies is its high heterogeneity. Indeed, HER2 is highly heterogeneous in GC[2,7–11] and others have shown that HER2 heterogeneity associates with resistance to HER2-targeted therapy[12–14]. Receptor endocytosis and recycling processes contribute to HER2 heterogeneity and membrane dynamics[15], affecting antibody-tumor binding and subsequent efficacy and antibody-dependent cytotoxicity (ADCC)-mediated mechanisms[16–25]. HER2 endocytosis occurs through caveolae[26], clathrin-[15] or endophilin-mediated mechanisms[27]. Caveolin-1 (CAV1), the major structural protein of cholesterol-rich caveolae, negatively correlates with membrane HER2 and affects Trastuzumab-tumor binding[18–24]. Endocytic trafficking systems also influence the efficacy of the antibody-drug conjugate (ADC) TDM1[17]. While CAV1-dependent endocytosis enhances cancer cells' chemosensitivity to TDM1[25], others have shown a role for caveolae-mediated endocytosis in TDM1 resistance[21,24]. Cholesterol-depleting drugs, statins, are FDA-approved drugs prescribed to millions of people worldwide for the treatment of hypercholesterolemia[28] and used in preclinical studies to modulate CAV1 protein levels[28–32]. In preclinical models, statins enhance HER2 confinement at the cell surface[18,33], increase HER2-directed immunoPET uptake and enhance Trastuzumab systemic efficacy in xenografts with non-predominant HER2 membrane staining[18].

In this work, we retrospectively validate CAV1 as a complementary biomarker for the selection of patients for HER2-targeted therapies. Tumors with high CAV1 correspond with low HER2 density at the cell surface and, in Trastuzumab trials, to patients with low survival rates. Using heterogeneous PDX models with varying levels of CAV1, we show that TDM1, an ADC that targets HER2, combined with lovastatin, a small molecule that depletes cholesterol in ways that modulate CAV1 protein expression, improves antibody-tumor binding and response rates better than either does alone. Mechanistically, statins enhance the disruption of downstream signaling and natural killer (NK) cells-mediated ADCC. Importantly, we validated these preclinical findings in retrospective analyses of patient-level data from clinical studies of HER2-targeted therapies in GC patients. It is possible that the findings herein reported are not limited to GC and should be considered while attempting to extend the clinical benefits of HER2-targeted therapies beyond breast and GC to other HER2-expressing solid tumors.

## Results

**CAV1-low profile predicts favorable GC response in patients undergoing Trastuzumab therapy.** Previous studies of HER2-positive tumor models implied a role for CAV1 in antibody binding and efficacy[18,20,21,24,25]. Our previous results of

### Table 1 Patient characteristics.

| Characteristics | N (%) |
|---|---|
| Total patients | 46 |
| Sex, median | |
| Male | 37 (80) |
| Female | 9 (20) |
| Age at diagnosis, median | 61 (between 28 and 87) |
| Therapy Prior to Trastuzumab | 9 (19.5) |
| EOX | 2 |
| Carboplatin/paclitaxel/RT | 2 |
| Carboplatin/taxol/RT | 1 |
| Modified DCF | 1 |
| FOLFOX | 2 |
| FOLFOX/regorafenib 2. 5-FU/ regorafenib | 1 |
| Stage of disease at the time of diagnosis | |
| Stage IV | 34 (74) |
| Stage III | 8 (17) |
| Stage II | 4 (9) |
| Stage of disease at the time of initiating Trastuzumab | |
| Stage IV | 46 (100) |
| HER2 IHC at diagnosis | |
| 3+ | 36 (78) |
| 2+ | 10 (22) |
| HER2-positivity retained after initiating Trastuzumab | 14 (30) |
| Sample type | |
| Primary | 20 (43) |
| Metastasis | 26 (57) |
| Sample location | |
| Liver | 12 (26) |
| Stomach | 10 (22) |
| Esophagus | 6 (13) |
| Esophagogastric junction | 6 (13) |
| Lung | 5 (11) |
| Brain | 4 (8.7) |
| Scalp | 1 (2.1) |
| Skin | 1 (2.1) |
| Peritoneum | 1 (2.1) |
| CAV1 IHC | |
| HER2+/CAV1^HIGH tumors | 12 (26) |
| HER2+/CAV1^LOW tumors | 34 (74) |
| Statin use | 19 (41) |
| Statin use prior Trastuzumab therapy | 12 (26) |
| Prior Trastuzumab at PDX biopsy | |
| Yes | 36 (78) |
| No | 10 (22) |
| Stage of disease at the time of PDX biopsy | |
| Stage IV | 40 (87) |
| Stage III | 6 (13) |

*EOX* chemotherapy combination of epirubicin, oxaliplatin, and capecitabine (Xeloda), *RT* radiation therapy, *Modified DCF* modified docetaxel, cisplatin, fluorouracil.

Trastuzumab in preclinical models of HER2-expressing xenografts[18] provided the rationale for a retrospective study stratifying CAV1-low and CAV1-high HER2+ GC. Eligible samples were obtained from previous trials at Memorial Sloan-Kettering Cancer Center (MSK) of HER2+ GC patients treated with Trastuzumab. Samples were from predominantly male patients (80% male versus 20% female) with a median age of 61 years (range 28–87). HER2-positivity was defined as IHC 3+, IHC 2+ and HER2:CEP17 FISH ratio ≥2.0, or *ERBB2* amplification by next-generation sequencing. Table 1 summarizes patient characteristics and Supplementary Fig. 1 shows patient survival in stratified HER2 IHC 2+ and 3+ tumors treated with Trastuzumab. The cohort consists of 46 patients with stage IV (74

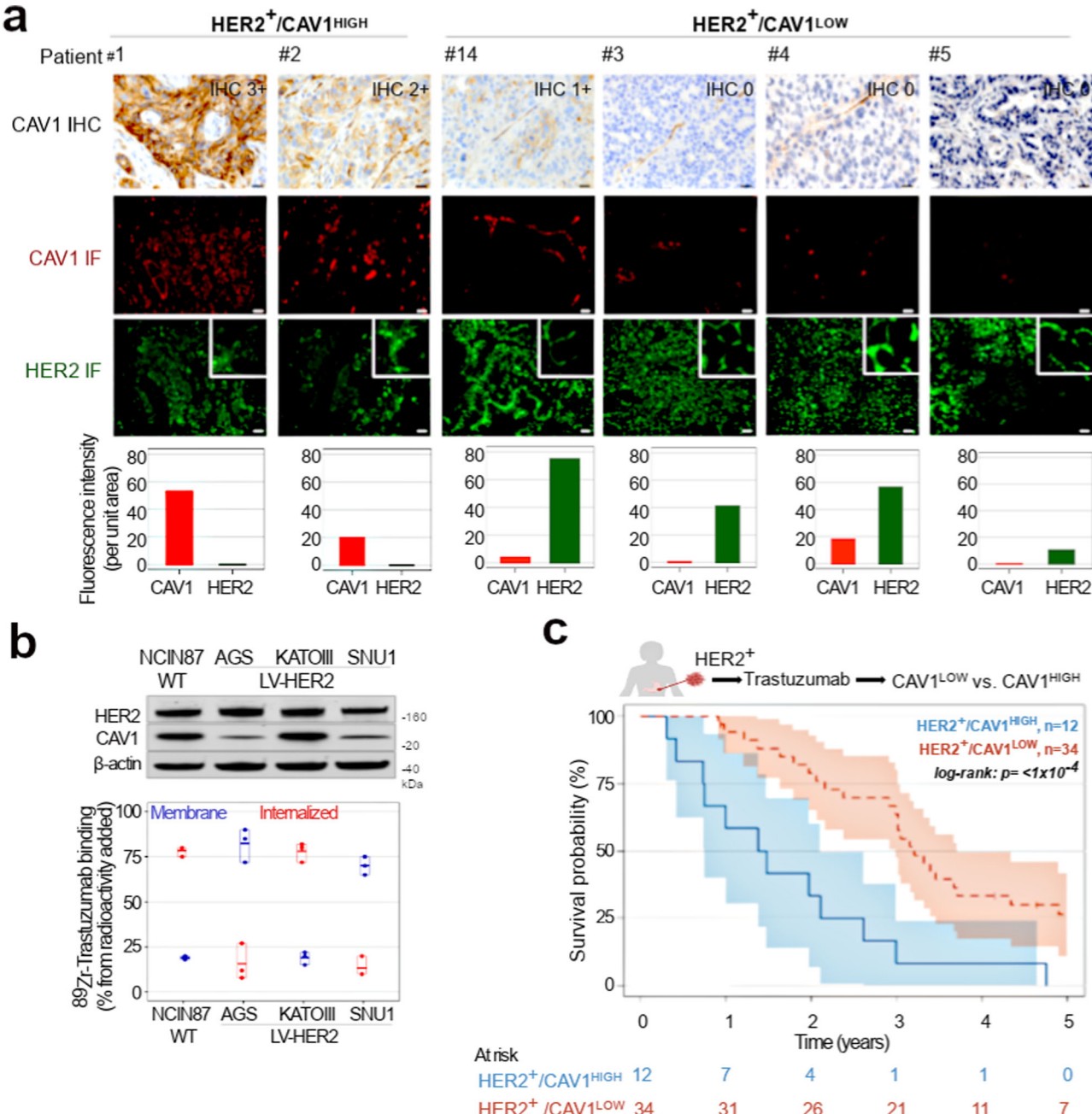

**Fig. 1 HER2 membrane levels and Trastuzumab efficacy depend on CAV1 protein levels. a** Immunohistochemical (IHC) detection and scoring intensity of CAV1, immunofluorescence (IF) staining of HER2 (green color) and CAV1 (red color) in HER2-expressing gastric tumor tissues. CAV1 reactivity at the cell membrane of tumor cells was considered for IHC scoring; IHC 0/1 + : CAV1-low (patient #14 and patients #3–5). IHC 2 + /3 + : CAV1-high (patient #1 and patient #2). The graphs plot protein fluorescence intensity per unit area, calculated by quantifying IF images (mean ± S.E.M, $n = 3$). Scale bar, 50 μm. HER2 membrane levels are classified as high versus low based on quantification of IF staining shown in Supplementary Fig. 5. Patient 1 to Patient 33 are IDs for all HER2+ gastric tumor tissues analyzed in the study (Supplementary Fig. 4). **b** $^{89}$Zr-labeled Trastuzumab (1 μCi, 0.25 μg) binding and internalization in NCIN87 GC cells wild-type (WT) and AGS, KATOIII, SNU1 GC sublines stably expressing HER2 (LV-HER2). **c** Kaplan–Meier analyses of CAV1 expression and GC disease outcome in patients treated with Trastuzumab. Patients with HER2+/CAV1$^{HIGH}$ (blue color, $n = 12$ patients) phenotype have a worse survival than HER2+/CAV1$^{LOW}$ (red, $n = 34$ patients). Log rank; $p < 1 \times 10^{-4}$. Source data are provided as a Source Data file.

%), stage III (17%), or stage II (9%) HER2+ GC disease at the time of diagnosis. All patients were stage IV at the point when Trastuzumab therapy was initiated. Samples obtained from patients enrolled on Trastuzumab trials (9/46 tumor samples were from patients that received other therapies prior to Trastuzumab) were analyzed for CAV1 IHC (Fig. 1a, b). This cohort was comprised of samples from primary tumors (43%) or metastases (57%). CAV1 IHC optimization used tissues with

varying levels of CAV1 (Supplementary Figs. 2 and 3). CAV1-staining at the membrane of GC was classified as 0/1 + CAV1-low (weak to low CAV1 membrane staining) and 2 + /3 + CAV1-high (moderate to strong CAV1 membrane staining; Fig. 1a, Supplementary Fig. 3). CAV1-high and CAV1-low IHC were detected respectively in 26% and 74% of HER2+ GC. In addition to CAV1 IHC, somatic alterations of patient samples used in our studies where determine by MSK-IMPACT (Supplementary

Fig. 4). This methodology uses a hybridization-based exon capture design to detect somatic single-nucleotide variants, small insertions and deletions, copy-number alterations, and structural rearrangements[10,34].

Immunofluorescence staining of patient samples revealed high membrane density of HER2 receptors in CAV1-low GC (Fig. 1a, Supplementary Fig. 5). Conversely, non-homogeneous HER2 membrane staining prevailed in CAV1-high tumors. Western blot studies of a panel of 6 GC cell lines supported the inverse correlation between HER2 and CAV1 protein expression (Supplementary Fig. 6A). These observations are consistent with previous results in preclinical models of HER2-expressing tumors[18–21].

To test Trastzumab binding in HER2$^+$ cancer cells expressing varying levels of CAV1, we first used a panel of GC cell lines (Fig. 1b, Supplementary Fig. 6A, B): the HER2$^+$ GC cell line (WT, NCIN87) and three GC cell lines (AGS, KATOIII, and SNU1) stably expressing HER2 (LV-HER2). In addition to the generation of KATOIII, AGS, and SNU1 sublines stably expressing HER2, we also attempted to express HER2 in MKN45 and SNU5 GC cells. Interestingly, the protocols herein used did not allow for the successful generation of LV-HER2 in cell lines containing the highest CAV1 expression (Supplementary Fig. 6B). Membrane-bound Trastzumab was higher in CAV1-low AGS LV-HER2 and SNU1 LV-HER2 GC cells when compared with CAV1-high NCIN87 or KATOIII LV-HER2 cells (Fig. 1b).

We then sought to determine if tumoral CAV1 was associated with survival in patients undergoing HER2-targeted therapy. We compared patient survival during Trastuzumab therapy in stratified CAV1-high and CAV1-low GC. These retrospective analyses used information about the 46 patients described above to determine the association of CAV1 IHC and HER2 membrane staining with Trastuzumab responses and overall survival in trials of HER2$^+$ GC (Table 1). In retrospective analyses performed in this study, the CAV1-low profile (34 of 46) corresponds to tumors with homogeneous surface receptors and predicts favorable patient response to Trastuzumab therapy (Fig. 1c, Supplementary Fig. 7).

**CAV1 depletion increases TDM1 binding in GC**. Membrane-localized receptors and trafficking are important in the therapeutic efficiency of ADCs[17]. We next hypothesized that differences in cell-surface HER2 in CAV1-high versus CAV1-low tumors (Fig. 1) results in different susceptibility to ADCs. To this end, we first established the significance of CAV1 levels on TDM1-tumor binding using HER2$^+$ gastric PDXs (78% of PDXs were obtained from patients prior initiating Trastuzumab; Table 1). The PDX tissues were confirmed to match the parent tissue shown in Fig. 1 by MSK-IMPACT data. Examination of H&E and IHC stained sections excluded the possible presence of B cell lymphomas in PDXs associated with Epstein–Barr Virus[35,36] (Supplementary Fig. 8; carcinomas: pancytokeratin$^+$/CD45$^-$/CD20$^-$, lymphomas: pancytokeratin$^-$/CD45$^+$/CD20$^+$). PDXs containing lymphoma were excluded from preclinical studies ($n = 13$). At 48 h post-injection of $^{89}$Zr-labeled TDM1, CAV1-low PDXs had uptakes ranging from $22.8 \pm 6.5$ to $32.5 \pm 5.7$ percentage of injected dose per gram of PDX (%ID/g), while CAV1-high PDXs yielded uptakes ranging from $9.7 \pm 3.6$ to $13.3 \pm 3.0$%ID/g (Fig. 2a). The lower TDM1 accumulation in CAV1-high xenografts, when compared with CAV1-low tumors, prompted us to interrogate if in vivo genetic depletion of CAV1 would boost ADC-tumor binding. To this end, we used CAV1-high NCIN87 GC cells containing incomplete HER2 surface density[18] to develop a Tet-On system of CAV1 knockdown in the presence of doxycycline (Dox); Supplementary Fig. 9A, B. Dox-induced CAV1 knockdown resulted in a 1.9-fold increase in HER2 at the cell membrane (Supplementary Fig. 9B). We performed in vivo studies in mice bearing

subcutaneous (s.c.) NCIN87 shRNA 486 or shRNA 479 xenografts. Control experiments included non-targeting control (NTC) shRNA xenografts. Mice were orally administered saline (OFF DOX) or Dox (ON DOX) for 11 days before $^{89}$Zr-labeled TDM1 injection (Fig. 2b). Transversal PET images of the saline cohort showed a gradual accretion of immunoPET signal between 24 and 72 h into the HER2-positive tumors (Fig. 2c). Antibody uptake was similar in OFF DOX and ON DOX cohorts of control shRNA NTC xenografts. On the other hand, xenografts of shRNA 486 or shRNA 479 showed a remarkably higher tumor uptake in ON DOX groups when compared with OFF DOX cohorts. Quantitation of the signal in tumors' regions of interest (ROI) further endorsed our findings from PET imaging (Fig. 2d). To temporally knockdown CAV1, ON/OFF Dox cohorts included mice treated with Dox for 7 days and saline for 4 days, (Supplementary Fig. 9C, D). The TDM1-tumor uptake using a ON/OFF Dox schedule was comparable in mice having NCIN87 shRNA NTC, shRNA 486, or shRNA 479 xenografts (Supplementary Fig. 9E). These results indicate that CAV1 knockdown enhances HER2 availability at the cell membrane resulting in an increase in TDM1-tumor binding in HER2$^+$/ CAV1$^{HIGH}$ NCIN87 xenografts.

**Statin-mediated CAV1 modulation is temporal and enhances TDM1 internalization**. Premised on our findings using the Tet-on system (Fig. 2c, d), we explored in vivo CAV1 depletion employing an FDA-approved pharmacologic approach with potential for clinical translation. Given that the cholesterol-depleting drug lovastatin modulates CAV1[18,37,38], we sought to determine whether lovastatin would enhance TDM1-tumor binding. TDM1 exhibits predominant surface localization after 1.5 h incubation time with lovastatin when compared with no-statin (Fig. 3a). The effect of statin-mediated TDM1-membrane binding is temporal and the ADC shows intracellular accumulation at 24 h incubation time with lovastatin. We next determined if differences in TDM1 binding would affect ADC internalization in GC cells. ADC internalization was measured using TDM1 conjugated with the pH-sensitive dye (pHrodo-TDM1) that only fluoresces in acidic environments, such as the lysosome[17]. We consistently observed that lovastatin increases TDM1 internalization in NCIN87, KATOIII LV-HER2, and SNU1 LV-HER2 GC cells (Fig. 3b), an effect that is rescued by the addition of mevalonate to the cell culture (Supplementary Fig. 10). Additional studies with $^{89}$Zr-labeled TDM1 demonstrate that lovastatin decreases ADC recycling to the cell membrane (Fig. 3b). TDM1 internalization in control and lovastatin-incubated cells co-localized with the lysosomal-associated membrane protein 1 (LAMP-1, Fig. 3c). Consistently, lovastatin does not alter the ubiquitination of immunoprecipitated HER2 (Supplementary Fig. 11). These data suggest that lovastatin enhances TDM1 binding to the surface of GC cells, which results in an increase in TDM1 internalization and decrease in ADC recycling.

The above data provided the rationale for preclinical imaging studies to explore the potential role of lovastatin as a CAV1 modulator in the context of TDM1 binding to HER2$^+$ GC. Mice bearing subcutaneous xenografts were orally administered the previously reported dose schedule of the cholesterol-depleting drug (two doses of 8.3 mg/kg given 12 h apart)[18]. Lovastatin induced a significant reduction in CAV1 tumor levels and increased HER2 membrane levels (Fig. 3d). At 48 h after the first dose of lovastatin, CAV1 expression and HER2 staining resembled those found at 0 h, lending further evidence for the transience and temporality of our statin regimen. To non-invasively monitor ADC uptake in statin cohorts, mice were intravenously injected with $^{89}$Zr-labeled TDM1 at 12 h after the first dose of lovastatin. The 12 h window for antibody injection

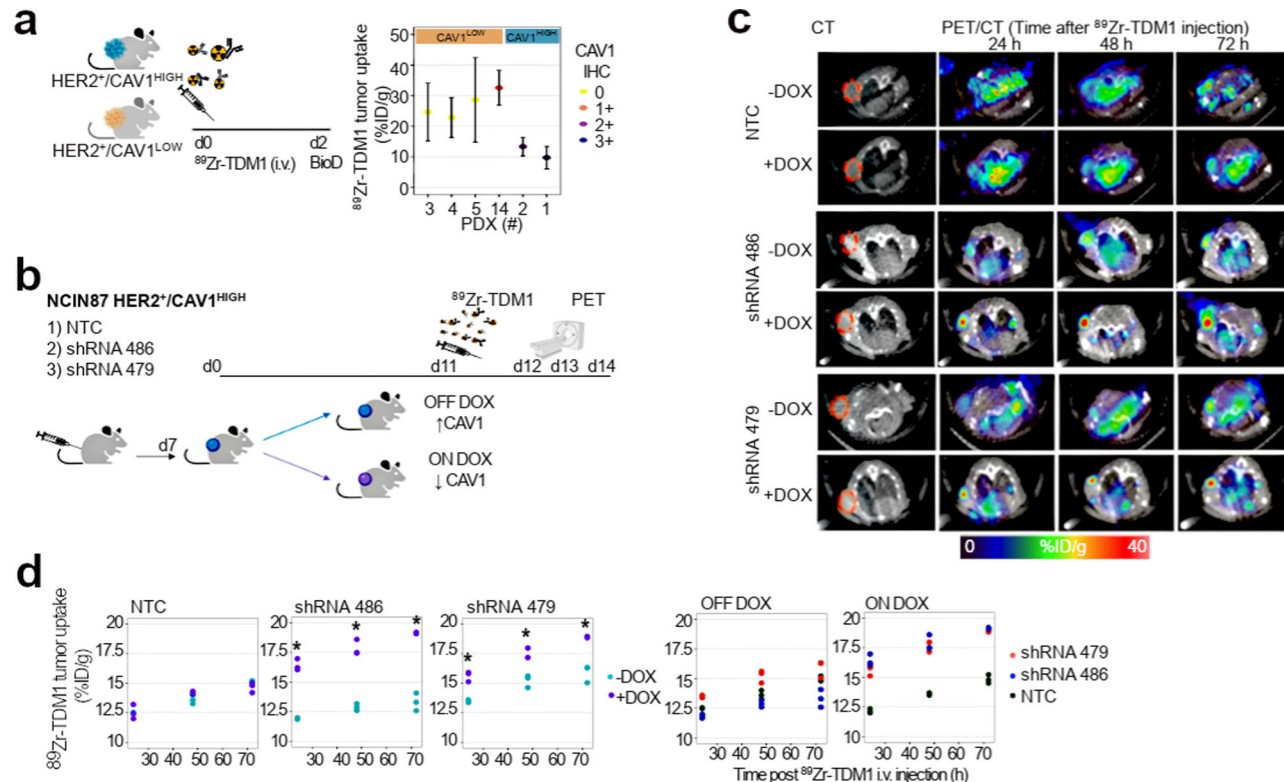

**Fig. 2 CAV1 depletion enhances TDM1-tumor binding. a** [$^{89}$Zr]Zr-DFO-TDM1 uptake in HER2-expressing gastric PDXs containing varying levels of CAV1. NSG mice bearing subcutaneous PDXs were intravenously administered with [$^{89}$Zr]Zr-DFO-TDM1 (6.66–7.4 Mbq, 45–50 μg protein) and biodistribution performed at 48 h p.i. of $^{89}$Zr-labeled antibody. PDX IDs in this figure match patient IDs shown in Fig. 1. Points, n = 5 mice per group, mean ± S.E.M. %ID/g, percentage of injected dose per gram. **b–d** Athymic nude mice bearing s.c. NCIN87 shRNA NTC, shRNA 486, or shRNA 479 xenografts were orally administered with 10 mg/mL of Dox (ON DOX) or PBS (OFF DOX) for 11 days. On day 11, mice were intravenously administered with [$^{89}$Zr]Zr-DFO-TDM1 (6.66–7.4 Mbq, 45–50 μg protein). PET images (**c**) were recorded at 24, 48, and 72 h p.i. [$^{89}$Zr]Zr-DFO-TDM1. The percentage of injected dose per gram (%ID/g) of TDM1 in tumors (**d**) was calculated by quantifying regions of interest (ROIs) in the PET images. *P < 0.05 based on a Student's t test, n = 3. Source data are provided as a Source Data file.

was based on our observations of CAV1 depletion and an enhancement in membrane HER2 (Fig. 3d). Control cohorts included mice orally administered saline instead of statin. The saline cohort revealed a radiopharmacologic profile standard for zirconium-89 labeled antibodies (Fig. 3e, Supplementary Fig. 12A) with gradual antibody accumulation to xenografts (4.9 ± 2.7, 10.2 ± 2.9, 22.2 ± 12.6, 31.1 ± 9.3 %ID/g at 4, 8, 24, and 48 h). However, the two doses of lovastatin yielded an antibody uptake higher at the different time-points when compared with the saline cohort (15.5 ± 9.7, 19.5 ± 9.3, 52.1 ± 7.6, 63.4 ± 16.7 %ID/g at 4, 8, 24, and 48 h). Oral administration of lovastatin results in images with high contrast and enhances tumor-to-background ratios (Fig. 3c, Supplementary Fig. 12B). To determine whether statin-mediated enhancement in ADC-tumor binding is dependent on CAV1 tumoral levels, we performed biodistribution studies with $^{89}$Zr-labeled TDM1 in the HER2-positive gastric PDXs shown in Fig. 1a. Although TDM1 accumulation in CAV1-low PDXs was similar in control and lovastatin cohorts (Fig. 3f), TDM1 uptake in PDX #1 (CAV1, IHC 3+) and PDX #2 (CAV1, IHC 2+) was 1.8-fold and 1.4-fold higher in lovastatin cohorts when compared with saline. These results indicate that acute CAV1 depletion by lovastatin increases cell surface receptors, enhancing TDM1 binding and internalization in HER2$^+$ PDXs.

**Statins enhance TDM1 efficacy.** To assess TDM1 efficacy in combination with lovastatin, we first conducted therapy studies in

CAV1-expressing HER2-positive (Fig. 3g–i) and HER2-negative GC cells (Supplementary Fig. 13A). In HER2-positive NCIN87 cells, TDM1 decreased viability (Fig. 3g), and lovastatin alone did not induce cell toxicity. However, statins greatly reduced cell viability when combined with the ADC. Of note, this effect was not observed in HER2-negative GC models (Supplementary Fig. 13B, C). In addition, cytotoxicity was significantly higher in TDM1/statin-treated cells than cells treated with Trastuzumab/statin (Fig. 3g). The increase in PARP cleavage further validated efficacy results with the combination therapy (Fig. 3h, i; Supplementary Fig. 13D, E). We next evaluated whether differences observed in cytotoxicity would interfere with HER2-mediated oncogenic signaling pathways. The phosphorylated proteins p-EGFR, p-ERK, and p-AKT were detected in both unstimulated and EGF-stimulated NCIN87 cells, suggesting that both MAPK and PI3K/AKT pathways are active (Fig. 3h, i; Supplementary Fig. 13D). TDM1 treatment alone in EGF-stimulated cells did not alter p-ERK, p-AKT, p-HER2, or p-HER3, in agreement with previous observations[39,40]. Lovastatin did not induce significant alterations in signaling, but when combined with TDM1 it reduces phosphorylation of both ERK and AKT in EGF-stimulated cells. Under heregulin (HRG) stimulation, the ADC decreases p-ERK and p-HER and, in combination with a statin, it effectively reduces p-ERK, p-AKT, and p-HER (Fig. 3h, i; Supplementary Fig. 13E). Collectively, these results show that statins enhance in vitro TDM1 efficacy by decreasing p-ERK and p-AKT oncogenic signaling pathways.

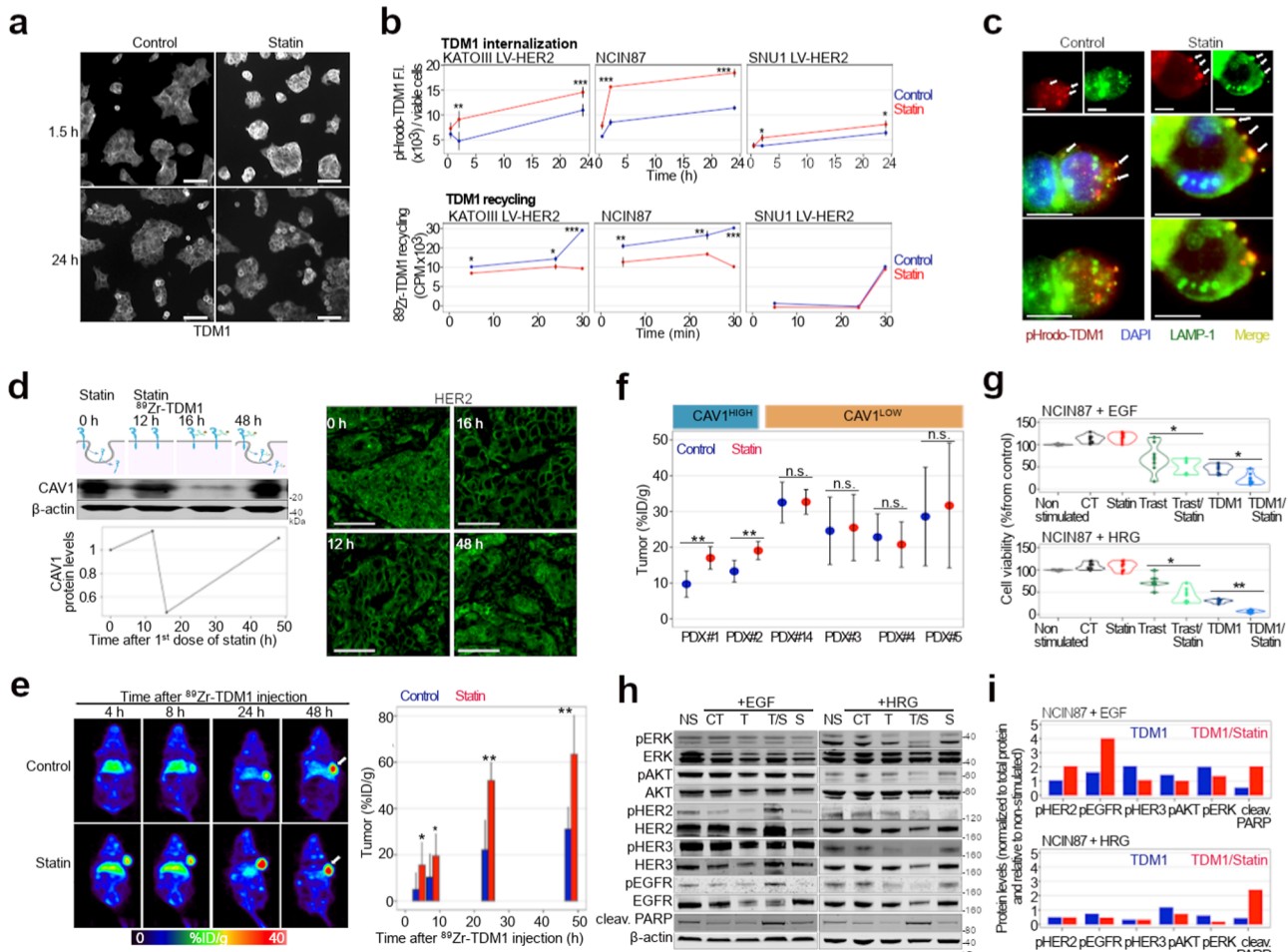

**Fig. 3 Statins enhance TDM1 binding and internalization. a** Confocal images of immunofluorescence of TDM1 in the presence or absence of lovastatin. Scale bars: 20 μm. **b** TDM1 internalization and recycling in NCIN87 GC cells wild-type (WT) and AGS, KATOIII, SNU1 GC sublines stably expressing HER2 (LV-HER2) in the presence and absence of lovastatin. The pHrodo-TDM1 fluorescent signal was normalized to the number of viable cells (*$P < 0.05$, **$P < 0.01$, ***$P < 0.001$ based on a Student's $t$ test, $n = 4$). **c** Confocal images of immunofluorescence staining of pHrodo-TDM1 and LAMP1 in NCIN87 cells in the presence and absence of lovastatin. Scale bars: 100 μm and 50 μm (inset). **d** Western blot of CAV1 and HER2 immunofluorescence in NCIN87 s.c. tumors from athymic nude mice. Lovastatin (8.3 mg/kg of mice) was orally administrated twice with an interval of 12 h between each administration. Scale bars: 50 μm. **e** Representative coronal PET images and TDM1-tumor uptake at 4, 8, 24, and 48 h p.i. of [89Zr]Zr-DFO-TDM1 in athymic nude mice bearing s.c. NCIN87 tumors. Lovastatin (8.3 mg/kg of mice) was orally administrated 12 h prior and at the same time as the tail vein injection of [89Zr]Zr-DFO-TDM1 (6.66–7.4 Mbq, 45–50 μg protein). Bars, $n = 5$ mice per group, mean ± S.E.M. *$P < 0.05$, **$P < 0.01$, ***$P < 0.001$ based on a Student's $t$ test. %ID/g, percentage of injected dose per gram. **f** [89Zr]Zr-DFO-TDM1 uptake in HER2-expressing gastric PDXs containing varying levels of CAV1 and administered saline (blue color) or statin (red color). PDX IDs in this figure match patient IDs shown in Fig. 1. Points, $n = 5$ mice per group, mean ± S.E.M, **$P < 0.01$ based on a Student's $t$ test. %ID/g, percentage of injected dose per gram. **g** Cell viability of NCIN87 cells at 48 h after cells incubation with Trastuzumab (Trast) and TDM1 alone or in combination with lovastatin. Bars, $n = 5$–7 per group, mean ± S.E.M. *$P < 0.05$, **$P < 0.01$, based on a Student's $t$ test. **h**, **i** Western blots of HER2 signaling and quantification of NCIN87 cells after 48 h incubation with TDM1 alone or in combination with lovastatin. Bars, quantification of Western blots shown in Fig. 3i. Supplementary Fig. 13 shows quantifications of three independent assays. Source data are provided as a Source Data file.

Encouraged by the in vitro cell death and signaling findings, we next determined TDM1 efficacy when combined with lovastatin using NCIN87 xenografts or PDXs shown in Fig. 1. Similar to the imaging studies reported above, therapeutic cohorts used PDXs obtained from patients prior initiating Trastuzumab therapy. Mice received intravenous injections of TDM1 (5 mg/kg once a week[39] for 5 weeks), oral doses of lovastatin (4.15 mg/kg administered 12 h prior and at the same time as the intravenous injection of antibody[18]), or a combination of ADC and lovastatin over 5 weeks (Fig. 4a). The vehicle and lovastatin cohorts had a similar trend of increased tumor volume over time (Fig. 4b). TDM1 alone inhibited tumor growth, but tumors developed resistance after 42 days of

therapy. The combination of the ADC with lovastatin greatly decreased tumor volume when compared with the monotherapy. In addition, TDM1/lovastatin decreases oncogenic signaling at 40 days after initiating therapies as we observed a reduction in p-AKT, p-ERK, and p-Tyr (Fig. 4c, Supplementary Fig. 13F). The phosphorylation of the cyclic (c)AMP responsive element binding protein (CREB), a player in HER2-mediated cancer development[41], was also lower in xenografts of mice treated with TDM1/lovastatin when compared with TDM1 alone (Fig. 4c, Supplementary Fig. 13F).

In addition to the conventional xenografts, PDX #1 was used to validate therapeutic studies (Fig. 4d, Supplementary Fig. 14). PDX #1 was obtained from HER2+ GC of a patient prior initiating

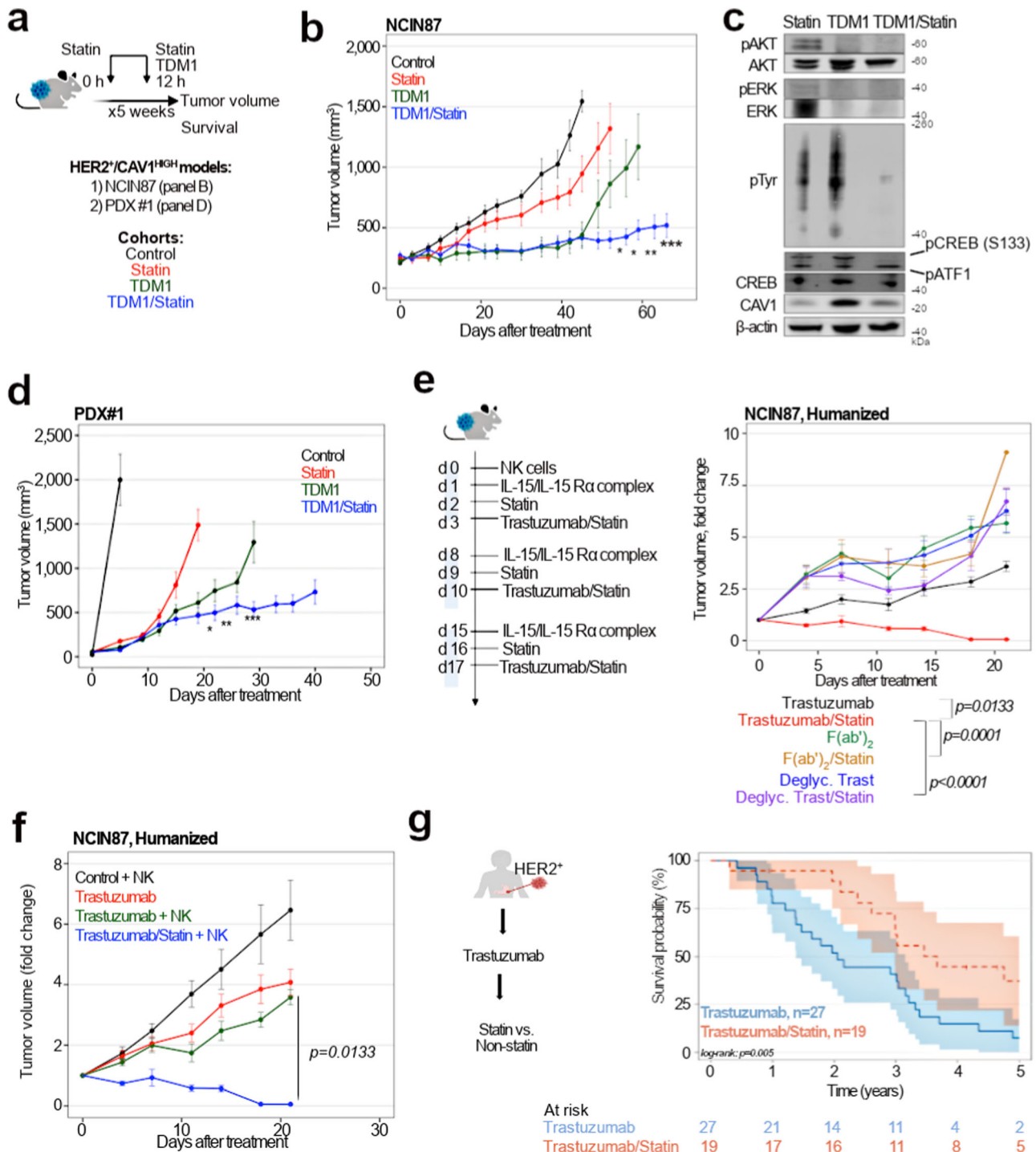

Trastuzumab therapy. Medical records indicated that this patient underwent first-line therapy with Trastuzumab but passed away less than a year after diagnosis. The patient developed brain metastases characterized by persistent *TP53* and *KRAS* somatic mutations, HER2 IHC 3+ and CAV1 IHC 3+. Lovastatin enhanced TDM1 efficacy in PDX #1 (Fig. 4d) which was accompanied by a decrease in p-ERK/p-AKT compared with monotherapy cohorts (Supplementary Fig. 15A). Notably, PDX #1 tumor volume in TDM1/lovastatin cohorts was higher than the previously reported Trastuzumab/lovastatin in the same GC PDX[18]. These preclinical results indicate that 2-weekly doses of statin (4.15 mg/kg) given over 5 weeks to mice with CAV1-high HER2[+] gastric xenografts enhance TDM1 efficacy.

**Statin enhances anti-HER2 antibody ADCC.** Receptor internalization affects ADC efficacy (Figs. 2–4) and diminishes antitumor immunity by ADCC[16], a major mechanism of clinical efficacy of IgG1 therapeutic antibodies. Although antibody/lovastatin delays tumor growth in immunodeficient mice via signaling inhibition, xenograft regrowth arises in immunodeficient hosts (Fig. 4b, d). Because Trastuzumab-mediated ADCC happens mainly through NK cells[42–44], we isolated NK cells from a healthy donor (Supplementary Fig. 16A–D) to measure ADCC in HER2[+] GC cells expressing different levels of CAV1 (Supplementary Fig. 6A, B). Lovastatin enhanced antibody ADCC in CAV1-positive GC cells but not in CAV1-negative cells (Supplementary Fig. 16E). Next, we conducted therapeutic studies in

**Fig. 4 Lovastatin enhances TDM1 efficacy and Trastuzumab-mediated ADCC. a–d** Superior in vivo therapeutic efficacy of TDM1 combined with lovastatin when compared with TDM1 alone. **a** Intravenous TDM1 administration 5 mg/kg weekly (for 5 weeks) was started at day 0. Lovastatin (4.15 mg/kg of mice) was orally administrated 12 h prior to and simultaneously with the intravenous injection of TDM1. Lovastatin enhanced TDM1 efficacy of $nu/nu$ female mice bearing NCIN87 gastric xenografts (**b**), and NSG mice bearing CAV1-high PDXs (**d**). $*P < 0.05$, $**P < 0.01$, $***P < 0.001$ based on a Student's $t$ test ($n = 8–10$ mice per group). **c** Western blot analyses of AKT, ERK, Tyr, CAV1, and CREB protein expression and phosphorylation in NCIN87 xenografts at 40 days after treatment with lovastatin, TDM1, or TDM1/lovastatin. **e** NSG mice bearing NCIN87 xenografts were intravenously injected $1 \times 10^6$ human NK cells at day 0. One day after NK cells tail vein injection, the IL-15/IL-15Rα complex was intraperitoneally administered at a dose of 1.25 µg/mouse. Trastuzumab or Trastuzumab/lovastatin efficacy was then evaluated during a cytokine-dependent NK expansion phase (week 1–week 3). Lovastatin enhanced Trastuzumab efficacy in NSG mice humanized with NK cells and bearing NCIN87 xenografts ($n = 8–10$ mice per group, mean ± S.E.M.). Statistical analyses performed using ANOVA coupled to Scheffé's method. **f** Trastuzumab/lovastatin efficacy is higher than the combination of Fc-silent Trastuzumab (Trastuzumab F(ab')₂ fragments or deglycosylated Trastuzumab) in NSG mice humanized with NK cells and bearing NCIN87 xenografts ($n = 8–10$ mice per group, mean ± S.E.M.). Statistical analyses performed using ANOVA coupled to Scheffé's method. **g** Kaplan–Meier analysis of statin use and HER2-expressing GC disease outcome in patients treated with Trastuzumab. Patients without statin treatment (blue color, $n = 27$) have a worse survival than patients treated with statin (red color, $n = 19$). Log rank; $p = 0.005$. Source data are provided as a Source Data file.

NK-humanized NSG mice bearing NCIN87 xenografts (Fig. 4e). The NK cohorts showed initial response to Trastuzumab, but tumor progression occurred on day 10 (Fig. 4e, Supplementary Fig. 16F). On the other hand, Trastuzumab/lovastatin combination therapy yielded tumor regression and stabilization. Control experiments used F(ab')₂ fragments and deglycosylated Trastuzumab (Supplementary Fig. 17) to remove the Fc-mediated stimulation of NK cell effector function. The combination of lovastatin with Fc-silent Trastuzumab shows lower efficacy than Trastuzumab/lovastatin (Fig. 4e).

To further confirm ADCC findings, we monitored Trastuzumab/lovastatin efficacy during cytokine-dependent (between week 1 and week 3) and cytokine withdrawal phases (weeks 4 to 5, Supplementary Fig. 18A). NK cells were present on day 7 after adoptive transfer, and NK expansion occurred during the cytokine-dependent phase (Supplementary Fig. 18B). The number of NK cells then decreased between days 21 and 30. All mice in the Trastuzumab/lovastatin cohort showed tumor regression and stabilization in tumor volume during the first 3 weeks (Fig. 4f, Supplementary Fig. 18C). In contrast, 7 out of 10 mice demonstrated tumor regrowth in the cytokine withdrawal phase. These results indicate that lovastatin enhances Trastuzumab efficacy, which depends on cytokine-mediated NK expansion and antibody's Fc domain.

**Patient survival is increased among statin users in Trastuzumab GC clinical trials.** Retrospective analyses of patients with GC undergoing Trastuzumab therapy further validated our preclinical findings (Fig. 4g). This study compared investigator-assessed responses between patients with and without background statin use for overall survival. Forty-one percent of patients (19 of 46) received statin as a background prescription while receiving Trastuzumab (Table 1). Twenty-six percent of patients were taking statins before initiating HER2-targeted treatments. Statin users were more frequently male (86% versus 14%) and more often older than 60 years of age (80% versus 20%). In the retrospective analyses, statin users treated with Trastuzumab had longer survival than non-statin users (log-rank, $p = 0.005$; Fig. 4g). Statin users in the CAV1-high expression group demonstrated longer survival when compared with non-statin users (log-rank, $p = 0.02$; Supplementary Figs. 19 and 20).

## Discussion

The retrospective clinical analyses and preclinical activity of antibody drugs, shown here, demonstrate that efficacy depends on density of the surface-localized receptors. These findings are significant in tumors with incomplete pattern of a membrane receptor for which antibody therapies are available. As an example, gastric tumors contain heterogeneous and dynamic levels of cell-surface HER2[2,7–9]. The variability in tumor response to antibody therapies among patients[5,6], all with seemingly HER2-expressing disease, suggests that patient selection should be optimized by incorporating other aspects of the biology. Tumors with high levels of tumor heterogeneity in HER2 expression have a poor response to TDM1[12–14]. CAV1 may be a complementary biomarker to detect receptor expression in the clinic and predictive biomarker of targeted therapeutic response as determined in our retrospective screening. This led to the hypothesis that acute CAV1 depletion is a potential pharmacologic strategy to anchor membrane receptors at the cell membrane, contributing to a more homogeneous receptor staining and further improving response to antibody therapy. From a mechanistic perspective, acute CAV1 depletion delays the recycling of ADC while enhancing disruption of downstream signaling and Fc-mediated stimulation of NK cell function.

CAV1 role as a tumor suppressor or promoter depends on the tumor type and disease stage[45]. In HER2-positive tumors, recombinant overexpression of CAV1 blocks oncogenic signaling[46]. In our studies, low tumoral levels of CAV1 correspond to tumors with high HER2 density at the membrane of GC and might be preferable when selecting patients for anti-HER2 antibody therapies. CAV1 modulates membrane levels of HER2 within the context of receptor endocytosis[18,22,23], interfering with the uptake and efficacy of antibodies or ADCs[18,20,21,24,25]. This study shows that the expression of CAV1 is an independent predictor of poor overall survival during Trastuzumab therapy in HER2⁺ GC.

Although TDM1 mechanisms of action are complex, the canonical model of ADC provides an helpful framework[47]: (i) ADC binds to the cell-surface receptor, (ii) ADC retains the antibody component and can decrease downstream signaling or induce ADCC, (iii) ADC-HER2 internalizes from the membrane into the intracellular compartment, and (iv) linker breakdown and DM1 drug release. Notably, the current status of patient selection for antibody-targeted therapies does not account for HER2 cellular distribution. It therefore may overestimate the amount of antigen available at the cell membrane for engaging the ADC. In this context, a lack of correlation between HER2 density and Trastuzumab accumulation in tumors is reported in clinical immunoPET imaging studies[12,48]. Others have shown that HER2 density at the cell membrane is a strong predictor of clinical outcome in patients with advanced breast cancer treated with Trastuzumab and chemotherapy[49]. Our immunoPET studies using PDX models that resemble the genetic complexity and heterogeneity of GC[50] indicate low TDM1 uptake in tumors containing high CAV1 protein levels. The impact of caveolae-mediated endocytosis has been reported for TDM1 and other ADCs: ADCs co-localize with CAV1 in resistant cancer cells[21,51].

However, it is important to keep in mind that a decreased receptor expression preventing the antibody drug from binding to GC is just one of many biological factors influencing antibody uptake. Other examples include truncated HER2 isoforms[52] or dysregulated mechanisms of ADC recycling, endocytosis, catabolism, and efflux of payload[15,17,21].

Kinase inhibitors[17,53,54] and CAV1 modulators[20] can temporally enhance HER2 membrane levels or promote ADC endocytosis in ways that improve TDM1 efficacy. While some reports suggest a positive role for CAV1 in promoting cells' sensitivity to TDM1[24,25], others have shown an association between resistance and caveolae-mediated ADC internalization[21]. CAV1 knockdown augments cell-surface HER2 half-life[18]; here, we confirm that CAV1 downregulation improves TDM1 binding to xenografts. In contrast, increased TDM1 efficacy occurs in vitro after enhancing CAV1 expression by metformin[20]. Although metformin is not a specific caveolae modulator and may exert other pleiotropic biologic effects, the proposed mechanism of action in that study is that caveolae mediate TDM1 endocytosis. Considering this previous report, it would seem disadvantageous to reduce CAV1 in tumors treated with TDM1 as it could decrease the ADC internalization. However, our data show that CAV1 modulation is unlikely to change HER2 degradation processes and ADC cellular trafficking. Instead, transient and controlled CAV1 depletion using statins boosts TDM1-tumor binding and internalization while reducing ADC recycling. This is consistent with previous preclinical studies temporally increasing cell-surface receptors to potentiate TDM1 therapy[54].

Controlled use of lovastatin, at doses lower than the maximum human dose, produce a temporal decrease in CAV1 protein levels[32]. Compared with lipophilic lovastatin, hydrophilic statins are less able to cross cancer cell membranes as they require active transport to enter cells[55]. Importantly, the lipophilic prodrug lovastatin depletes tumoral CAV1 in ways that improve antibody binding to tumors[18,31,32]. Similarly, lovastatin enhances TDM1 uptake in CAV1-high PDXs, confirming an association between the loss of CAV1 and ADC uptake. Additionally, statins do not alter TDM1 accumulation in tumors with low levels of CAV1. Still, a large number of other proteins involved in non-caveolae internalization pathways are temporally affected by statins[32]. Nonetheless, specific genetic depletion of CAV1 in vivo enhances ADC accumulation in tumors. Thus, these data show that acute depletion of CAV1, whether induced by synthetic oligonucleotides or small molecules, is an important event for anti-HER2 antibody binding to GC.

After reaching tumor cells and engaging with membrane HER2, TDM1 retains the functionality of its naked antibody, Trastuzumab. Therefore, the initial therapeutic mechanisms of TDM1 start before ADC internalization and payload release inside the cancer cells. From a mechanistic perspective, lovastatin enhances the Fab-mediated activity of Trastuzumab responsible for disrupting oncogenic signaling. When compared with Trastuzumab, statins induce higher ADC therapy, suggesting they ultimately increase DM1-mediated cytotoxicity. However, temporally enhancing cell-surface HER2 overall may not augment therapeutic outcomes if other resistance mechanisms (e.g., alterations in oncogenic signaling and tumors' vulnerability to microtubule-directed chemotherapy) are at play[47]. In addition to the Fab region, the Fc portion of the antibody increases cell death by orchestrating ADCC. We show that CAV1 depletion boosts NK-mediated ADCC, an important mechanism of the clinical effectiveness of Trastuzumab[42–44]. Others have also shown that endocytosis modulation enhances Fc-mediated ADCC of antibodies against HER or PD-L1[16]. The increased ADCC in combination strategies using statin depends

on HER2 surface levels, the Fc-domain of the antibody, and the presence of cytokine-expanded NK cells. In addition to the direct role of statins in modulating intratumoral cholesterol, future studies are necessary to determine the specific direct and indirect mechanisms of statins in enhancing antibody efficacy. Other studies have demonstrated that statins also act as immunomodulatory drugs[56]. Therefore, future studies are necessary to determine whether the statin doses necessary for modulation of cell-surface HER2 induce alterations in immune cells including NK cells.

In vitro lovastatin concentrations reported here are higher than nanomolar range concentrations detected in patient prescriptions. Therefore, in vitro results herein described might not translate to clinical dosages. In preclinical models, lovastatin was administered on a two-dose schedule. Therefore, low and variable amounts of the cholesterol-depleting drug will accumulate and reduce tumoral CAV1. However, our retrospective findings of statin use during patient treatment with Trastuzumab uphold the translational relevance of results obtained in PDXs. Note that these analyzes collected data from patients receiving standard statin doses for cardiovascular indications, while our therapeutic studies relate to mice with normal cholesterol levels. Additionally, the patient survival analyses do not account for variables that possibly cause residual confounding observations (e.g., economic status including screening and access to medical care, physical activity, obesity, and diet). Further, the ability of statins to increase Trastuzumab efficacy may be a result of the cholesterol lowering itself (hepatic actions of statins) or may depend on the tumor's genotype, and future analyses in increased sample size are necessary to determine factors of statin-induced efficacy in combination with Trastuzumab. Although a prospective trial is needed to confirm this combination approach fully, our preclinical and retrospective studies show statins' potential to enhance antibody-directed therapies in GC.

An extensive pharmacodynamic/pharmacokinetic study is required for a clinical investigation of statin use in combination with HER2-targeted therapies. Additional studies are also necessary to determine whether patients with GC on statins have a prolonged survival time, regardless of HER2 status or Trastuzumab treatment. Although previous studies have shown that statins have antitumor effects in vitro and in preclinical models, few studies have explored the relationship between statin use and the survival of patients after cancer treatments. The outcome of these studies will grant insights into possible coincidental consequences on non-tumor cells, unsought toxicities, and statin doses for clinical use. Based on the medical records, statin users included here initiated statins before or while on Trastuzumab therapy and presumably tolerated these cholesterol-depleting agents well. Other studies also suggest that statins prevent heart failure in patients receiving Trastuzumab[57]. Therefore, these results indicate that the combination approach may hold a clinically acceptable safety profile and may achieve reasonable tumors selectivity when used in a controlled manner.

In summary, CAV1 may serve as a predictive biomarker when selecting tumors for HER2-targeted therapies. Importantly, immunoPET allows measurements of differences in Trastuzumab or ADC binding to CAV1-high versus CAV1-low tumors. Statin-mediated temporal increase in HER2 receptors at the cell surface has the capability to enhance Trastuzumab and TDM1 efficacy. Our findings can potentially be extended to other antibody therapies and tumor types characterized by heterogeneous patterns of receptors at the cell membrane of tumor cells. These studies may help guide future clinical trials into integrating statins —forthwith available, well-tolerated, and affordable agents—for combination approaches in cancer treatment.

## Methods

**Ethical compliance**. This study was approved by the Memorial Sloan Kettering Cancer Center (MSKCC) Institutional Review Board (IRB). All patient-derived xenografts (PDXs) were generated by the Antitumor Assessment Core at MSKCC (IRB biospecimen protocol #14-091, PI: de Stanchina). Patients informed consent was required and obtained for all cases (IRB# 06-107, IRB#12-245, NCT02954536, IRB# 06-103, NCT01913639, NCT01522768). Animal studies were performed in the MSKCC animal facility in compliance with institutional guidelines under Institutional Animal Care and Use Committee (IACUC) approved protocols (MSKCC No. 08-07-013, PI: Lewis). Human NK cells were obtained from STEMCELL Technologies - Institutional Biosafety Committee # LAB201900146 (PI: Lewis).

**HER2 positivity, MSK-IMPACT assay, and retrospective data**. HER2$^+$ (HER2 IHC 3+, HER2 IHC 2+ and HER2:CEP17 FISH ratio ≥ 2.0 or *ERBB2*-amplified by next-generation sequencing) gastric tumor tissues were obtained from Trastuzumab trials at MSK led by Y.J. (NCT02954536, IRB# 06-103, NCT01913639, NCT01522768)[10]. Patient clinical information was collected manually from the electronic medical record (M.M. and M.L.). The presence of somatic alterations in HER2-expressing tumors was analyzed by MSK-IMPACT[10,58].

**CAV1 immunohistochemistry (IHC) optimization and scoring**. CAV1 IHC optimization was performed by the Laboratory of Comparative Pathology at MSK. Anti-CAV1 antibodies obtained from two different commercial sources (Abcam, ab2910 and BD Biosciences, 610407) were used for IHC. After comparing CAV1 IHC in various human tissues, the BD Biosciences antibody demonstrated lower unspecific reactivity than the Abcam antibody. IHC staining for CAV1 was performed on formalin-fixed paraffin-embedded HER2-expressing gastric tumor sections (5 µm thickness) on a Leica Bond RX automated stainer (Leica Biosystems, Buffalo Grove, IL). Following deparaffinization and heat-induced epitope retrieval in a citrate buffer at pH 6.0, the primary antibody against CAV1 (BD Bioscience, 610407) was applied at a concentration of 1:250 (v/v). A polymer detection system that includes an anti-mouse secondary antibody (DS9800, Novocastra Bond Polymer Refine Detection, Leica Biosystems) was then applied in the tumor samples. The 3,3′-diaminobenzidine tetrachloride was used as the chromogen, and the sections were counterstained with hematoxylin.

Initial titration studies of CAV1 IHC optimization used human tonsil tissues (Supplementary Fig. 2). After CAV1 immunohistochemical validation, both anti-CAV1 Abcam ab2910 and BD Biosciences 610407 were used to stain a HER2$^+$/CAV1$^{LOW}$ and HER2$^+$/CAV1$^{HIGH}$ tumor expresser (Supplementary Fig. 2). The high and low CAV1 tumor expressers were based on previous IF analyses (Supplementary Fig. 5). The anti-CAV1 antibody from BD Biosciences demonstrated low unspecific binding (Supplementary Fig. 2). The CAV1 IHC scoring was performed by a board-certified veterinary pathologist (S.M.). The pathologist performed a blind histopathological examination without prior knowledge of CAV1 IF or HER2 membrane levels. The slides were scored according to the standard IHC scoring for HER2 in human tumors. Only CAV1 reactivity associated with the membrane of neoplastic cells was considered for scoring. Cytoplasmic CAV1 reactivity in neoplastic cells, endothelial CAV1 reactivity in stromal blood vessels (Supplementary Fig. 2B), or CAV1 non-specific reactivity in necrotic regions (Supplementary Fig. 2C) was ignored for IHC scoring. Samples with CAV1 IHC 0/1+ and CAV1 IHC 2 + /3+ were classified as CAV1-low and CAV1-high, respectively.

**Immunofluorescence (IF) staining of HER2 and CAV1 in tumor tissues**. The MSK Molecular Cytology Core Facility performed CAV1 and HER2 immunofluorescence staining of formalin-fixed, paraffin-embedded sections (10 µM) sections. The whole slide digital images of HER2 and CAV1 staining were obtained on Pannoramic MIDI scanner (3DHistech, Hungary) at a resolution of 0.3250 µm per pixel. Regions of interest around the cells were then drawn and exported as.tif files from these scans using Caseviewer (3DHistech, Hungary.) These images were then analyzed using ImageJ/FIJI (NIH, USA) to measure fluorescence intensity after applying a median filter and background subtraction.

**Cell lines, cell culture, and treatments with lovastatin**. Human GC cell lines NCIN87, AGS, SNU5, SNU1, and KATOIII, were purchased from American Type Culture Collection (ATCC). MKN45 GC cells, embryonic kidney 293 cells (HEK 293), and 293FT cells were gifts from the Rudin Lab and Weisser Lab at MSK. All cell lines were mycoplasma-free and cultured at 37 °C in a humidified atmosphere at 5% CO2 until a maximum passage of 15. The MSK integrated genomics operation core performed cell line authentication using short tandem repeat analysis.

All cell culture media were supplemented with 100 units/mL penicillin and streptomycin. NCIN87 GC cells were maintained in Roswell Park Memorial Institute (RPMI)-1640 growth medium supplemented with 10% fetal calf serum (FCS), 2 mM L-glutamine, 10 mM hydroxyethyl piperazineethanesulfonic acid (HEPES), 1 mM sodium pyruvate, 4.5 g/L glucose and 1.5 g/L sodium bicarbonate. KATOIII cells were grown in Iscove's Modified Dulbecco Medium (IMDM) growth medium supplemented with 20% FCS and 1.5 g/L sodium bicarbonate.

MKN45 cells were kept in RPMI-1640 supplemented with 2 mM L-glutamine. SNU1 cells were maintained in RPMI-1640 containing 10% FCS. AGS cells were grown in Kaighn's Modification of Ham's F-12 Medium (F-12K Medium) supplemented with 10% FCS. SNU5 cells were grown in IMDM containing 20% FCS.

For in vitro experiments with lovastatin, cells were incubated with 25 µM of the active form of lovastatin (Millipore) for 4 h prior addition of Trastuzumab or TDM1[18].

**Generation of GC lines stably expressing HER2 (LV-HER2)**. The pHAGE-*ERBB2* (Addgene plasmid 116734) was a gift from Gordon Mills and Kenneth Scott; lentiviral envelope and packaging plasmids pMD2.G (Addgene plasmid 12259) and psPAX2 (Addgene plasmid 12260) were gifts from Didier Trono. The plasmids were purified using QIAquick Spin Miniprep or Plasmid Plus Midi kits (Qiagen) and verified by Sanger sequencing (Genewiz) before lentiviral production. Lentivirus was produced by transfection of HEK293T cells using the JetPrime system (Polyplus). The ratio of pMD2.G:psPAX2:pHAGE-*ERBB2* was 1:2:3, the ratio of JetPrime transfection reagent to DNA was 2:1, and the ratio of JetPrime buffer:transfection reagent was 50:1. The HEK293T cells were incubated with the DNA and transfection reagents for 24 h before the media was changed. Two days after replacing the media, the media (herein referred to as viral supernatant) was collected and filtered through 0.45 µM PVDF filters (Millipore). The viral supernatant was then concentrated 20-fold with Lenti-X Concentrator (Clontech) according to the manufacturer's instructions. The GC cell lines KATOIII, MKN45, SNU5, AGS, and SNU1 were transduced using 8 µg/mL hexadimethrine bromide (Sigma), and the media was changed 24 h after transduction. Three days after transduction, puromycin selection of HER2-expressing cells was initiated on all cell lines at concentrations from 1 to 2.5 µg/mL, and selection was continued for at least 4 days. The overall increase in HER2 cellular expression was validated by Western blot (Supplementary Fig. 6).

**Preparation of human NK cells**. An 81 mL leukapheresis pack containing $5.50 \times 10^9$ white cells with a viability of 98% was shipped at ambient temperature from STEMCELL Technologies (Supplementary Fig. 16A) and processed immediately upon receipt. On the day of arrival, the COVID-19 PCR result was pending for the donor. Therefore, samples were handled following Biosafety guidelines at MSK for human samples of unknown COVID-19 status of source case. Because the procedures were non-aerosol generating, the samples were handled as BSL2. Upon removing the leukapheresis sample was washed by adding 81 mL of EasySep buffer (20144, STEMCELL). The sample was then centrifuged at $500 \times g$ for 10 min at room temperature. Upon removal of the supernatant, the cell pellet was resuspended in EasySep buffer at $5.50 \times 10^7$ cells/mL. An ELISA test for COVID-19 was performed before isolation of NK cells using the KT-1032 EDI$^{TM}$ Novel Coronavirus COVID-19 Elisa kit. After confirming that the sample was COVID-19 negative, the NK cells were isolated by negative selection using the human NK cell enrichment kit (19055, STEM Cell). Briefly, the enrichment cocktail (50 µL/mL) was added to the sample containing $5.50 \times 10^7$ cells/mL and incubated for 10 min at room temperature. The magnetic particles (100 µL/mL) were then added and incubated for 10 min at room temperature. The sample was placed in the EasySep magnet for 10 min. The isolated NK cells were then transferred into a new tube and the NK cell population was confirmed by FACs as CD3$^-$CD56$^+$ cells (Supplementary Information 16C, D).

**Flow cytometry**. After NK cell isolation, cells were washed twice with ice-cold PBS. NK cells were then split into groups and stained with APC-hCD56 (clone HCD56, 318309, Biolegend) and PE-hCD3 (clone HIT3a, 300307, Biolegend). After 20 min of incubation, NK cells were washed with PBS containing 2% (v/v) FBS and fixed in 4% paraformaldehyde (PFA). NK cells were then resuspended in FACS buffer (PBS containing 2% FBS and 2 mM EDTA) and placed on ice until analysis. Single color controls were made, NK cells were identified as CD3$^-$CD56$^+$, and results were analyzed with Flowjo software (Flowjo LLC v10.7.1). Flow cytometry was performed in the MACSQuant Analyzer 10.

**Western blots**. Whole-protein extracts from cells or tumors were prepared after cell scrapping or tissue homogenization, respectively, in RIPA buffer and separated on SDS-PAGE gels (NuPAGE 4–12% Bis-Tris Protein Gels, Invitrogen). Membranes were probed using the following primary antibodies: rabbit anti-CAV1 1:500 (Abcam, ab2910), rabbit anti-HER2 1:800 (Abcam, ab131490), mouse anti β-actin 1:20,000 (Sigma, A1978), rabbit anti-ubiquitin 1:1,000 (Cell Signaling Technology, 3933 S), mouse anti-ERK 1:100 (Invitrogen, 14-9108-80), rabbit anti-pERK 1:500 (Invitrogen, 700012), rabbit anti-AKT 1:1,000 (Cell Signaling Technology, 9272 S), rabbit anti-pAKT, 1:2,000 (Cell Signaling Technology, 4060 S), rabbit anti-cleaved PARP, 1:1,000 (Cell Signaling Technology, 9541 S), rabbit anti-pHER2, 1:500 (Abcam, ab53290), rabbit anti-HER3, 1:500 (Abcam, ab32121), rabbit anti-pHER3, 1:2,500 (Abcam, ab76469), rabbit anti-EGFR 1:1,000 (Abcam, ab52894), rabbit anti-pEGFR 1:500 (Abcam, ab40815), mouse anti-pTyr 0.5 µg/mL (EMD Millipore, 05-321X), rabbit anti-CREB 1:1,000 (Cell Signaling Technology, 9197 S), rabbit anti-pCREB 1:1,000 (Cell Signaling Technology, 9198 S).

The membranes were then incubated with secondary antibodies IRDye®800CW anti-rabbit or anti-mouse IgG 1:15,000 (LI-COR Biosciences) and imaged on the Odyssey Infrared Imaging System (LI-COR Biosciences) followed by densitometric analysis.

**PathScan antibody array kit**. The MAPK Phosphorylation Antibody Array (Abcam, ab211061) was used to determine MAPK signaling changes. Total tissue lysates (500 µg) were loaded in the membranes according to the manufacturer's instructions. The membrane arrays were then incubated with the detection antibody cocktail, and the HRP-Anti-Rabbit IgG was used to detect bound proteins. The proteins were visualized using the detection buffer mixture on a chemiluminescent blot documentation system consisting of x-ray film with a film processor followed by densitometric analysis.

**HER2 immunoprecipitation**. NCIN87 cancer cells were incubated in media with 5% (v/v) of FBS in the presence of 10 µM of the proteasome inhibitor MG-132 (Sigma-Aldrich). Cells were incubated with 10 µg/mL of TDM1 in the presence and absence of lovastatin at 37 °C for 4 h. Cells were then washed with cold PBS and lysed with NP-40 buffer (150 mmol/L NaCl, 10 mmol/L Tris pH 8, 1% NP-40, 10% glycerol). Forty micrograms of proteins were used as total lysates. For immunoprecipitation, protein lysates (500 µL of NP-40 buffer containing 200 µg of protein) were incubated with 10 µg of primary antibody Neu (F-11) agarose conjugate (sc-7301; Santa Cruz Biotechnology) overnight at 4 °C with gentle rotation. The pellet containing the immunoprecipitated fraction was collected by centrifugation at $1000 \times g$ for 30 s at 4 °C, washed three times with NP-40 buffer and once using nuclease-free sterile water before resuspension in Laemmli buffer.

**Cell viability and HER2 signaling analyses**. Cell viability was determined in cells treated with Trastuzumab, TDM1, Trastuzumab/lovastatin, or TDM1/lovastatin. Cells were plated in a 96-well plate ($1 \times 10^4$ cells/well) and pre-cultured for 24 h. Cells stimulated with 100 ng/mL of EGF or HRG were incubated with 20 nM of Tastuzumab or TDM1 in the absence or presence of lovastatin. Cell viability was measured at 48 h after treatments using thiazolyl blue tetrazolium bromide (MTT, Sigma). The optical density value was read at 570 nm using the Spectra Max ID5 (Molecular Devices). The percentage of cell viability was indicated by comparison with cells in the absence of stimulation or treatments.

In Western blot assays of HER2 signaling, cells were plated in a six-well plate (1 million cells/well). The day after, cells stimulated with EGF or HRG were incubated with Trastuzumab, TDM1, Trastuzumab/lovastatin, or TDM1/lovastatin. Total cell extracts were collected 48 h after cells' treatment and analyzed by Western blot.

**In vitro therapeutic ADCC**. Cells were plated at a 50:1 effector (NK):target (GC) ratio in serum-free cell culture medium supplemented with 0.1% BSA. Cells were treated with 100 µg/mL of Trastuzumab or TDM1 in the absence or presence of lovastatin. After 6 h of incubation time, cell death was measured by determining LDH release using the Cytotoxicity Detection Kit (LDH; Roche).

**Tetracycline-inducible shRNA CAV1 expression (Tet-On system)**. A panel of 5 different shRNA against CAV1 and 1 NTC shRNA were generated by the Gene Editing & Screening Core at MSK and cloned into the LT3GENIR4(pRRL) vector. This backbone contains a neomycin selection and an inducible Dox system (Tet-On, Supplementary Fig. 9). The viral particles were produced using ExtremeGene HP (Roche) and 293FT packaging cells using a 3rd generation lentivector packaging system (3 vector system). The NCIN87 cells were then infected for 24 h. After 24 h, fresh media was added to the cells. After 48 h from the infection, antibody selection was initiated at 1200 µg/ml of neomycin and kept for 2 weeks. The cells were then placed on Dox for 48 h at a concentration of 1 µg/mL to induce GFP expression and CAV1 knockdown before sorting for GFP. Dox was removed from the media, and cells were expanded for 10 days in the absence of Dox (to return CAV1 expression to baseline levels and diminish GFP expression). The overall decrease in CAV1 expression after cells incubation with Dox was validated by Western blot (Supplementary Fig. 9).

**Conjugation and radiolabeling of TDM1 or Trastuzumab**. Trastuzumab or TDM1 were obtained from the MSK Hospital Pharmacy. The pHrodo-TDM1 was obtained by conjugating the free lysine residues of TDM1 with the amine-reactive pH-sensitive pHrodo iFL Red STP ester dye (ThermoFisher Scientific, P36014) according to the manufacturer's instructions.

To prepare [⁸⁹Zr]Zr-DFO-antibody, TDM1 or Trastuzumab were first conjugated with the bifunctional chelate *p*-isothiocyanatobenzyl-desferrioxamine (DFO-Bz-NCS; Macrocyclics, Inc) and then labeled with zirconium-89 (⁸⁹Zr)[59]. Radiochemical purity (RCP) was determined by instant thin-layer chromatography. The radiolabeled conjugates used for in vitro and in vivo studies had a RCP of 99%, radiochemical yields ranging from 92 to 97%, specific activities in the range of 21.98–24.73 Mbq/nmol, and immunoreactivities above 90%.

**Binding, internalization, and recycling assays**. For the binding assays, solutions of ⁸⁹Zr-labeled Trastuzumab or TDM1 (4 µCi/µg) were prepared in PBS (pH 7.5) containing 1% w/v human serum albumin (HSA, Sigma) and 0.1% w/v sodium azide (NaN₃, Acros Organics). Cells (1 million) were incubated with 1 µCi (0.25 µg) of the radiolabeled antibody for 1 h at 4 °C. Unbound radioactivity was removed, and cells were washed twice with PBS by centrifugation. The pellet-bound radioactivity was measured on a gamma counter calibrated for zirconium-89.

For internalization assays, cells were plated in a 96-well plate (50,000 cells/well). The day after, cells were incubated with 5 nM of pHrodoTDM1 for 30 min at 4 °C. Cells were then incubated at 37 °C, and fluorescent measurements were performed between 30 min and 24 h hours after incubation with pHrodo-TDM1. Fluorescence was recorded using Spectra Max ID5 (Molecular Devices) at excitation wavelength 560 nm/emission wavelength 585 nm. Cell viability was determined using the MTT assay, and the pHrodo-TDM1 fluorescent signal was normalized to the number of viable cells at each time point. To determine whether mevalonic acid treatment rescued the lovastatin effect, the cell lines were treated with lovastatin and 200 µM of R-Mevalonic Acid (Santa Cruz Biotechnology) for 4 h.

For recycling assays, cells were plated in a 6-well plate (1 million cells/well). The day after, cells were incubated with 1 µCi (0.25 µg) of the radiolabeled antibody in media at 37 °C. After 4 h incubation time, cells were kept in ice, washed with ice-cold PBS, and the supernatant was collected. Cell surface-bound radiotracer was collected by cells incubation at 4 °C for 5 min in 0.2 M glycine buffer containing 0.15 M NaCl, 4 M urea at pH 2.5. The cells were then incubated in media at 37 °C to allow recycling processes. Antibody recycling to the cell membrane was measured at 5, 25, and 30 min after washing cells and collecting the cell surface-bound radiotracer. The radioactive fractions were measured for radioactivity on a gamma counter calibrated for zirconium-89.

**Immunofluorescence assays of TDM1**. For immunofluorescence assays, cells were plated at 0.1 million cells/slide in chamber slides (154526, ThermoFisher Scientific) for 24 h. Cells were then incubated with 1 µM TDM1 for 90 min or 24 h at 37 °C. Cells were fixed with 4% PFA, permeabilized with 1% Triton X-100 in PBS (pH 7.4) and blocked with 5% bovine serum albumin in PBS buffer, before incubation with the DAPI and secondary goat anti-human IgG fluorescently labeled with Alexa Fluor 488 (A-11013, ThermoFisher Scientific).

For immunofluorescence assays of GC cells with pHrodo-TDM1 and LAMP-1, cells grown in chamber slides were incubated with 1 µg/mL of pHrodo-TDM1 for 48 h. After cells fixation with PFA and permeabilization using 1% Triton X-100, cells were incubated with a rabbit anti-LAMP-1 primary antibody (ab24170, Abcam). Cells were then incubated with DAPI and secondary goat anti-rabbit IgG fluorescently labeled with Alexa Fluor 488 (A-11008, ThermoFisher Scientific).

**Antibody deglycosylation and F(ab')2 fragments generation**. Trastuzumab deglycosylation was achieved by adding 1.1 units of recombinant PNGaseF enzyme (New England BioLabs) per 1 µg of antibody. Trastuzumab (3 mg, 144 µL) was mixed with 3000 units of PNGaseF enzyme (13. 3 µL from a stock solution containing 225 U/µL), 25 µL 500 mM sodium phosphate (pH 7.5), and 31.7 µL of water. The reaction was incubated at 37 °C for 2 h. To remove the PNGaseF enzyme from the reaction mixture and purify deglycosylated Trastuzumab, chitin magnetic beads (100 µL, E8036S, New England Biolabs) were added to the reaction mixture.

The F(ab')₂ fragments were generated using Trastuzumab and the F(ab')₂ fragmentation kit following the manufacturer's instructions (G-Biosciences).

**Tumor xenografts and patient-derived xenografts (PDXs)**. The experimentation involving animals followed the guidelines approved by the Research Animal Resource Center and IACUC at MSK (New York, NY), the ARRIVE guidelines, and the guidelines for the welfare and use of animals in cancer research. The maximum allowed total tumor burden of 2 cm³ was not exceed in our experiments.

NCIN87, NCIN87 shRNA NTC, NCIN87 shRNA 486, or NCIN87 shRNA 479 cancer cells were subcutaneously implanted in female athymic nude mice *nu/nu* (8–10 weeks old, Charles River Laboratories). A total of 5 million cells were suspended in 150 µL of a 1:1 v/v mixture of medium with reconstituted basement membrane (BD Matrigel, BD Biosciences) and injected subcutaneously in each mouse.

PDX models were established by the Anti-tumor Assessment Core, from tumor specimens collected under an approved institutional review board protocol by the Research Animal Resource Center and IACUC at MSK, NY[50]. Briefly, tumors were minced, mixed with Matrigel, and implanted subcutaneously in 6–8-week-old NSG mice (Jackson Laboratories). PDXs used in imaging and therapeutic experiments were obtained from patients prior initiating Trastuzumab therapy.

The tumor volume (V/mm³) was estimated by external vernier caliper measurements[18].

**PDX genetic and immunohistochemical validation**. To confirm that PDXs herein used recapitulate parent tissue, MSK-IMPACT data were obtained in both PDX and human tumor tissues. Given that patient-derived EBV-positive lymphomas are often observed in PDX models using NSG mice[35,36], H&E and IHC stained sides were reviewed by a board-certified veterinary pathologist (S.M.) to exclude lymphomas in PDX models. IHC was performed by the Laboratory of

Comparative Pathology at MSK for pancytokeratin, (primary antibody Dako Z0622 applied at 1:500 concentration), human CD45 (Dako M0701, 1:100), and human CD20 (Dako M0755, 1:1000) on the Leica Bond RX automated staining as described above for the CAV1 IHC method. Carcinomas were confirmed by IHC as pancytokeratin⁺/CD45⁻/CD20⁻. B cell lymphomas excluded from preclinical studies ($n = 13$) were IHC pancytokeratin⁻/CD45⁺/CD20⁺.

**CAV1 modulation using genetic and pharmacologic approaches.** For preclinical imaging studies using the Tet-On system, mice were randomly assigned into the following groups ($n = 5$ mice per group): *OFF DOX*, daily oral administration of PBS for 11 days prior to tail vein injection of ⁸⁹Zr-labeled TDM1; *ON DOX*, daily oral administration of 10 mg/mL of Dox for 11 days prior to tail vein injection of ⁸⁹Zr-labeled TDM1; *ON/OFF DOX*, daily oral administration of 10 mg/mL of Dox for 7 days followed by oral administration of PBS for 4 days before tail vein injection of ⁸⁹Zr-labeled antibody.

For preclinical imaging studies using lovastatin, mice were assigned into the following groups ($n = 5$ mice per group)[18,32]: *Control*, oral administration of PBS 12 h prior to and at the same time as the tail vein injection of ⁸⁹Zr-labeled TDM1; *Lovastatin*, oral administration of lovastatin (8.3 mg/kg of mice) 12 h prior to and at the same time as the tail vein injection of ⁸⁹Zr-labeled TDM1.

**Small-animal PET and acute biodistribution studies.** Mice bearing subcutaneous xenografts or PDXs (100–150 mm³ in tumor volume) were randomized before administering [⁸⁹Zr]Zr-DFO-TDM1 (6.66–7.4 Mbq, 45–50 μg protein) by tail vein injection. PET imaging ($n = 3$ mice per group) and ex vivo biodistribution ($n = 5$ mice per group) were performed according to previously reported methods[18,31,32]. PET images were analyzed using ASIPro VM software (Concorde Microsystems). Radioactivity present in each organ was expressed as the percentage of injected dose per gram of organ (% ID/g).

**In vivo therapeutic efficacy.** Mice with subcutaneous xenografts or PDXs of volume between 100 and 300 mm³ were randomly grouped into treatment cohorts ($n = 10$ per group): control, TDM1, Trastuzumab, lovastatin, TDM1/lovastatin, or Trastuzumab/lovastatin. Mice received weekly intravenous injections of TDM1 (5 mg/kg) or intraperitoneal injections of Trastuzumab (5 mg/kg) for 5 weeks. Lovastatin (4.15 mg/kg of mice) was orally administered 12 h prior and at the same time as the intravenous injection of TDM1. Tumor volumes were determined twice a week.

**In vivo therapeutic ADCC.** NCIN87 GC cells (5 million cells suspended in 150 μL of a 1:1 v/v mixture of medium with reconstituted basement membrane) were subcutaneously implanted in female severely immunodeficient NSG (6–8 weeks old, Jackson Laboratories). Once NCIN87 GC tumor volumes reached 100–150 mm³, freshly isolated NK cells (1 million cells in 200 μL PBS) were administered by tail vein injection. The interleukin-15/ interleukin-15 receptor alpha complex (IL-15/IL-15Rα complex) was used to achieve NK cell expansion and activation in vivo[60,61]. One day after NK cells tail vein injection and once per week, the IL-15/IL-15Rα complex was intraperitoneally administered at a dose of 1.25 μg/mouse. Mice were randomly grouped into treatment cohorts ($n = 10$ per group): saline, lovastatin, Trastuzumab, Trastuzumab/lovastatin. Mice received weekly intraperitoneal injections of Trastuzumab (5 mg/kg). Lovastatin (4.15 mg/kg of mice) was orally administered 12 h prior and at the same time as the intraperitoneal injection of Trastuzumab. Control cohorts included treatments in NSG mice that were not intravenously administered NK cells. Additional control experiments were performed using Fc silent deglycosylated Trastuzumab and Trastuzumab F(ab′)₂ fragments. Tumor volumes were determined twice a week.

**Quantification, statistical analyses, and reproducibility.** Data were analyzed using R v3.6.0. (http://www.rstudio.com/) or GraphPad Prism 7.00 (www.graphpad.com). Statistical differences between mean values were determined using analysis of variances (ANOVA) coupled to Scheffé's method or a Student's $t$ test. To compare treatments between cell lines, the Wilcoxon–Mann–Whitney test was performed using a 1-sided alpha of 0.05. The overall patient survival is defined as the time from diagnosis to death. Patients alive are censored at their date of last follow-up. Survival rates are estimated using Kaplan–Meier estimator, and curves are compared using the log-rank test. Data shown for western blot analyses represents three independent experiments.

**Reporting summary.** Further information on research design is available in the Nature Research Reporting Summary linked to this article.

## Data availability
All data generated or analyzed during this study are included in this published article (and its Supplementary Information Files). Source data are provided with this paper.

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

## Acknowledgements

We acknowledge the MSK Small-Animal Imaging Core Facility, the Radiochemistry and Molecular Imaging Probe Core, the Biostatistics Core, the Anti-tumor Assessment Core and Molecular Cytology Core Facility and the Immune Monitoring Facility, which were supported by NIH grant P30 CA08748. This study was supported in part by the Geoffrey Beene Cancer Research Center of MSK, NIH NCI R35 CA232130, NIH R01 CA244233-01A1. We gratefully acknowledge Mr. William H. and Mrs. Alice Goodwin and the Commonwealth Foundation for Cancer Research and The Center for Experimental Therapeutics of MSK. P.M.R. Pereira acknowledges the Tow Foundation Postdoctoral Fellowship from the MSK Center for Molecular Imaging and Nanotechnology, the Alan and Sandra Gerry Metastasis and Tumor Ecosystems Center of MSK, the American Cancer Society (IRG-21-133-64-03), and NIH (R01 CA244233-01A1). We gratefully acknowledge Dr. Ricardo D'Oliveira Albanus from Department of Computational Medicine & Bioinformatics, University of Michigan for assistance in RStudio analyses. We would also like to acknowledge Dr. Marco Russo and Daniel Zakheim from the Gene Editing & Screening Core at MSK to assist in the Tet-on system. We are grateful to Dr. Elisa De Stanchina and all the team at the Antitumor Assessment Core for helping with the PDX models. We thank Dr. Fiona Simpson and Dr. Joseph Sun insightful suggestions regarding the experiments with NK cells. We are also thankful to Dr. Monica Shooken and Dr. Luis Batista for letting us use the BioTek plate reader and the Odyssey Infrared Imaging System, respectively.

## Author contributions

*Conception and design:* P.M.R.P., J.S.L. *Development of methodology:* P.M.R.P., K.M., S.M., M.C., S.S.P., A.K.T., A.R., L.K., M.M., Y.Y.J. *Acquisition of data (provided animals, acquired and managed patients, provided facilities, etc.):* P.M.R.P., K.M., S.M., M.L., K.T., M.C., S.S.P., A.M., A.R., L.K., M.M., Y.Y.J., J.S.L. *Analysis and interpretation of data (e.g., statistical analysis, biostatistics, computational analysis):* P.M.R.P., K.M., S.M., M.L., K.T., M.C., S.S.P., A.M., A.R., L.K., M.M., Y.Y.J., J.S.L. *Writing, review, and/or revision of the paper:* M.R.P., K.M., S.M., M.L., K.T., M.C., S.S.P., A.M., A.R., L.K., M.M., Y.Y.J., J.S.L. *Administrative, technical, or material support (i.e., reporting or organizing data, constructing databases):* P.M.R.P., K.M., S.M., M.L., K.T., M.C., S.S.P., A.M., A.R., L.K., M.M., Y.Y.J., J.S.L. *Study supervision:* M.R.P., Y.Y.J., J.S.L.

## Competing interests

The authors declare no competing interests.
