## [Peer Review File · Nature Communications]

Reviewers' comments:

Reviewer #1 (Remarks to the Author):

This report describes a retrospective clinical study analysis and a PDX and cell-based study examining the abundance of cell surface HER2 and cell surface caveolin 1 in gastric cancer tumors from MSKCC, and the subsequent response of the patients to trastuzumab. The goal was to establish that caveolin 1 could predict response to HER2 targeted agents in gastric cancer. This manuscript includes a lot of work and likely represents years of effort.

Specific comments:

1. Lines 165-166: Mention the sample sizes (7 and 20) of tumors upon which this statement is based.
2. Figure 2B and Figure 3G: Are these PDXs available to investigators? Are the numbers meaningful?
3. Lines 265-266: Qualify this statement to indicate that the experiment was done with a single cell line. Results from a single cell line are generally considered not publishable.
4. Figure 3G: Check the axis: Time after statin incubation (h)?
5. Figure 4: This figure has a worrisome level of duplication of data. In Figures 4 A, D and F remove the duplicated data. Use the lines with error bars and remove the dot plots.
6. Line 339 and Figure 4D: This experiment needs a control group. The text state that survival was improved; however, the endpoint seems to be tumor volume – not survival.
7. Figure 4I and Table 1: Lovastatin has been tested in cancer settings many times as a modulator/enhancer of other therapeutics. It is very likely that numerous gastric cancer patients on lovastatin or other cholesterol lowering drug have also received trastuzumab or TDM, perhaps the journey should continue. The data shown on Table 1 and Figure 4I on this point is an intriguing lead.
8. Line 485: Change the incorrect word dose to the correct word 'concentration' in both places.
9. Line 511: Find a more credible word to describe immunopET than 'incredible'.

Reviewer #2 (Remarks to the Author):

The reason pertuzumab or TDM1 failed may have had nothing to do with CAV1 expression (compared to these drugs succeeded in breast cancer). There is considerable differences in Her2 biology in gastric cancer and breast cancer, particularly. Do you have data to show CAV1 levels in breast cancer and whether there is any correlation with Her2 expression, survival, and benefit from the two drugs mention. If not, then the issue may be more complicated than suggested.

The use of TDM1 (in stead of trastuzumab) complicates interpretation. TDM1 (you often focus on internalization, which is important for its action but may have nothing to do with CAV1) has a cytotoxic moiety. The experiments would have been easy interpret by using trastuzumab and not an ADCC.

The retrospective patient population also limits interpretation. Several different chem-regimens were used. This basically cannot be normalized. Introduces bias which cannot be sorted out.

Introduction can be reduced by 50%.

Patient survival (in the retrospective cohort) should also be shown simply by Her2 3+ or 2+ expression.

Patient population was treated with trastuzumab (and different chemo regimens) but most experiments were done with TDM-1. Complicates interpretation.

The IMPACT panel focused on Her2 only and other data are shown in a limited manner. Not clear. the number of patients with NGS is rather small. Not sure what value this analysis has provided.

Majority of PDXs were pretreated with trastuzumab.

Did you check CAV1 expression heterogeneity like it occurs with Her2 expression. This can be accomplished easily by checking 2-3 patient samples from proximity. If there is considerable heterogeneity in expression (meaning 2+/3+ in one sample and negative in another) it also complicates results and makes conclusion unsteady.

Back to the Impact panel results, mutations are not frequent in the entire population like this. Not sure what the message is. No conclusions can be drawn from this small sample size.

You suggest that CAV1 low tumors had different signaling profile than CAV1 high tumors, however, the approach is not unbiased. unless a more comprehensive interrogation is carried out (meaning we don't know what is happening with other pathways we don't check), we cannot be certain.

Many results are correlative. Mechanism results are not shown.

Her2 expression (in the context of CAV1) and internalization (which has its own unique set of circumstances) are confusing results.

CAV1 depletion experiments are best done with trastuzumab (not TDM1). Page 9

Correlation of CAV1 expression (should be in conjunction of Her2 expression) should be considered with trastuzumab and not TDM1. Page 9

Conclusions (Suppl Fig 5E) that CAV1 KO enhances TDM-1 binding. This is very confusing. Was Her2 expression increased to promote TDM-1 binding? Please clarify Page 10.

Additionally, Fig 3B and Suppl Fig 6A focus on internalization of TDM-1. Please provide data on trastuzumab and Her2 expression. Also provide data on CAV1 as modulated by lovastatin. Serum lipids changes would be nice and complementary.

Statements around line 266 (page 11) about internalization (no change) are also confusing and need clarification.

Reviewer #1:

This report describes a retrospective clinical study analysis and a PDX and cell-based study examining the abundance of cell surface HER2 and cell surface caveolin 1 in gastric cancer tumors from MSKCC, and the subsequent response of the patients to trastuzumab. The goal was to establish that caveolin 1 could predict response to HER2 targeted agents in gastric cancer. This manuscript includes a lot of work and likely represents years of effort.

Response: We are very grateful to the reviewer for the positive overview of the manuscript and constructive suggestions to improve the paper. We feel that with these comments the paper has been significantly improved and impact has increased.

Specific comments:

1. Lines 165-166: Mention the sample sizes (7 and 20) of tumors upon which this statement is based.

Response: Taking these comments into consideration and following Reviewer #1 and Reviewer #2 suggestions, we have removed the comparisons of IMPACT data in CAV1-low versus CAV1-high tumors. We agree that it does not seem to add much to your manuscript, mainly since there are no significant differences between the CAV1 high/low groups except for TMB, and it is not clear whether this is meaningful. We have ongoing studies that aim at exploring genetic somatic alterations in HER2-positive CAV1-high versus CAV1-low breast, gastric, and lung tumors. All of these studies in robust sample size would be necessary to determine significant differences in CAV1-high and -low tumors.

2. Figure 2B and Figure 3G: Are these PDXs available to investigators? Are the numbers meaningful?

Response: The Antitumor Assessment core at MSKCC, led by Dr. Elisa de Stanchina (destance@mskcc.org) establishes, characterizes, and provides PDXs of different cancer models. Interested investigators in using the PDXs should reach out to Dr. de Stanchina and/or the corresponding author of the study reporting the PDX of interest. PDXs are available upon an MTA, as well as approval of changes in IRB for each request.

To the reviewer's second question, Figure 2B and Figure 3G are now Figure 2A and Figure 3F. In our studies, we used PDXs that were obtained from patients prior to initiating trastuzumab therapy that successfully grew when implanted in a mouse. Although we initially tested a higher number of PDX models, we did not include those in the present manuscript for four main reasons: 1) they did not grow in a mouse, 2) they had uncontrolled growth in mice which would introduce bias in our studies as high tumor volumes would result in unspecific antibody-tumor binding

as a result of EPR effects, 3) IMPACT data in the PDX demonstrated differences when compared with the patient tissues from which it was originated, and 4) several of the PDXs demonstrated to be lymphomas instead of carcinomas as determined during our quality control assessment of PDXs. Therefore, to avoid confounding factors in our analyzes, we only report these 6 different PDXs that contain varying levels of CAV1. To determine the sample size in each one of our cohorts, Dr. Mauguen (8th author in the manuscript) who is an Assistant Attending Biostatistician performed the following statistical considerations:

Based on preliminary data, the average biodistribution is 9 %ID/g in the control group (sd=5), varying from 2.6% ID/g at 4h to 17.5 %ID/g at 48h. To conclude that it is 50% higher in the lovastatin group (mean = 13.5 %ID/g), with a power of 80% and a 2-sided alpha of 0.05, we need a total of 5 mice per group. With this sample size, we had a 76% power to show a significant interaction with time.

Therefore, we used 5 mice per group in our biodistribution studies with xenograft models.

3. Lines 265-266: Qualify this statement to indicate that the experiment was done with a single cell line. Results from a single cell line are generally considered not publishable.

Response: We thank the reviewer for noticing this and pointing it out for us to make a correction. We have now included additional cancer cell lines to demonstrate that by enhancing HER2 availability at the cell surface, lovastatin increases TDM1 internalization and delays TDM1 recycling. Our new data includes TDM1 internalization and recycling assays in NCIN87, KATOIII LV-HER2, SNU1 LV-HER2 gastric cancer cell lines (kindly see the new **Fig. 3B**).

Page 9: “Using the pHrodo-TDM1 conjugate, we consistently observed that lovastatin increases TDM1 internalization in NCIN87, KATOIII LV-HER2, and SNU1 LV-HER2 GC cells (Fig. 3B). Additional studies with 89Zr-labeled TDM1 demonstrate that lovastatin decreases ADC recycling to the cell membrane (Fig. 3B).”

4. Figure 3G: Check the axis: Time after statin incubation (h)?

Response: Figure 3G is now Figure 3F. The axis labels are correct.

The yy axis shows TDM1 accumulation (%ID per gram of tumor tissue) in HER2-positive gastric PDXs CAV1-high (PDX #1 and PDX #2) and CAV1-low (PDX #14, #3, #4, #5).

The xx axis relates to the different PDXs studied in the presence (red color) or absence (blue color) of lovastatin.

As described in the legend, the different PDXs were administered the same schedule and dose of lovastatin: “Lovastatin (8.3 mg/kg of mice) was orally

administrated 12 h prior and at the same time as the tail vein injection of [⁸⁹Zr]Zr-DFO-TDM1”.

5. *Figure 4: This figure has a worrisome level of duplication of data. In Figures 4 A, D and F remove the duplicated data. Use the lines with error bars and remove the dot plots.*

Response: In our therapeutic experiments, we have plotted tumor volume over time in separate graphs for each cohort because tumor growth and endpoints are different for each mouse in the same group. As suggested by the reviewer, we have modified Figure 4 to only show average data with error bars. However, we did not remove the dot plots from the manuscript but moved those to Supplementary Information to have our data published with the highest transparency (**Supplementary Fig. 12**). We believe that the dot plots further capture the heterogeneous response of tumors to the treatment.

6. *Line 339 and Figure 4D: This experiment needs a control group. The text state that survival was improved; however, the endpoint seems to be tumor volume – not survival.*

Response: We regret the oversight. Following this comment, we have updated Figure 4D to include tumor growth of the control cohort. We have removed the survival graph from Figure 4D and modified the wording in the manuscript to avoid misunderstandings.

Page 11: “Lovastatin enhanced TDM1 efficacy in PDX #1 (Fig. 4D) which was accompanied by a decrease in p-ERK/p-AKT compared with monotherapy cohorts (Supplementary Fig. 13A).”

7. *Figure 4I and Table 1: Lovastatin has been tested in cancer settings many times as a modulator/enhancer of other therapeutics. It is very likely that numerous gastric cancer patients on lovastatin or other cholesterol lowering drug have also received trastuzumab or TDM, perhaps the journey should continue. The data shown on Table 1 and Figure 4I on this point is an intriguing lead.*

Response: We appreciate the reviewer’s positive feedback about future studies in the context of using cholesterol-lowering drugs to enhance antibody therapies. We believe that the suggested alterations and experiments have highly increased the quality of our manuscript and strengthened the conclusions of our study.

8. *Line 485: Change the incorrect word dose to the correct word ‘concentration’ in both places.*

9. *Line 511: Find a more credible word to describe immunoPET than ‘incredible’.*

Response: All the minor points were accepted and included in the revised version of the manuscript.

Reviewer #2

The reason pertuzumab or TDM1 failed may have had nothing to do with CAV1 expression (compared to these drugs succeeded in breast cancer). There is considerable differences in Her2 biology in gastric cancer and breast cancer, particularly. Do you have data to show CAV1 levels in breast cancer and whether there is any correlation with Her2 expression, survival, and benefit from the two drugs mention. If not, then the issue may be more complicated than suggested.

Response: We most sincerely regret the misunderstanding. The manuscript has been carefully reviewed to ensure clarity and avoid any misunderstanding or lack of our appreciation for the differences in HER2 biology in different tumor types. Kindly see below examples of changes made to our manuscript:

Page 4: “Although several targeting agents are effective in treating HER2-positive breast tumors [1, 2], not all tumors benefit from HER2-targeted therapies (reviewed in [2]) due to considerable differences in HER2 biology in different tumor types.”

Page 15: “However, it is important to keep in mind that a decreased receptor expression preventing the antibody drug from binding to GC is just one of many biological factors influencing antibody uptake. Other examples include truncated HER2 isoforms [54] or dysregulated mechanisms of ADC recycling, endocytosis, catabolism, and efflux of payload [15, 17, 21].”

Although our present study describes the role of caveolin-1 in temporal regulation of HER2 trafficking and therapeutic efficacy of trastuzumab-DM1 conjugate in HER2-positive gastric cancers, our current data *does not support assumptions or conclusions on the role of CAV1 for the failure of pertuzumab or TDM1 in patients with gastric cancer*. We have ongoing studies in HER2-positive gastric, breast, and lung tumors that aim at determining the contributions of CAV1 for HER2 heterogeneity in these tumors.

To the reviewer’s second half of this critique, our previously reported preclinical data (Pereira et al. Nat Commun 2018) showed that CAV1-high cancer cells contain low levels of HER2 available at the cell membrane, that caveolin-1 knockdown or depletion with statin increases HER2 at the cell membrane and increases trastuzumab efficacy in different tumor models including BT474 breast xenografts. In addition to these previously reported preclinical studies and expanding our research in breast cancer, we are currently conducting retrospective analyses on a cohort of HER2-positive metastatic breast cancer patients treated at MSKCC who received T-DM1. *Our preliminary data in breast cancer is showing a median progression-free survival of 14 months in statin users compared to 5.4 months in those who had no record of statin use*. Additionally, we are performing CAV1 IHC in these breast cancer samples to determine differences in patient response to Trastuzumab or TDM1. This work is extensive given the different collaborators involved, the need for optimization of the experiment’s protocols, as well as the

increased sample size. We would respectfully suggest that our ongoing investigations in breast cancer and other HER-positive tumors belong to another paper.

We sincerely believe and hope that the reviewer will find the revised form of our manuscript suitable for publication in Nature Communications. To merit that status, we have made an earnest attempt to address the reviewer's comments and have extensively revised the manuscript. Please read further.

The use of TDM1 (instead of trastuzumab) complicates interpretation. TDM1 (you often focus on internalization, which is important for its action but may have nothing to do with CAV1) has a cytotoxic moiety. The experiments would have been easy interpret by using trastuzumab and not an ADCC.

Response: We most respectfully disagree with this reviewer's critique. We believe that our work presents compelling experimental evidence on the role of CAV1 in regulating HER2 endocytic traffic of oncogenic receptors, which have a key role in ADC internalization and efficacy. In our previous publication (Pereira et al. Nat Commun 2018), we showed the involvement of CAV1 protein in HER2 cell membrane dynamics in the context of receptor endocytosis. Several of the experiments requested by this reviewer in the context of Trastuzumab are already reported in our previous publication or shown in the present manuscript. The reason why our retrospective clinical data was performed with Trastuzumab and not with TDM1 is that we do not have patient data available for TDM1 in GC since it failed in clinical trials to treat patients with gastric cancer.

In the present manuscript, we took on a line of experiments to show the important role of CAV1 in tumors response to TDM1 efficacy: 1) CAV1 knockdown increases surface HER2 and improves antibody binding in vitro and in vivo; 2) use of lovastatin as a tool for CAV1 temporary suppression and increased antibody binding; 3) an increase in TDM1 binding as mediated by statin enhances ADC internalization and delays recycling; 4) improved ADCC mediated by NK cells due to increased surface latency of anti-HER2 antibody and antibody-FcR interaction.

The retrospective patient population also limits interpretation. Several different chem-regimens were used. This basically cannot be normalized. Introduces bias which cannot be sorted out.

Response: We do agree with the reviewer's suggestion that additional retrospective studies excluding patients that received chemotherapy need to be included in our analyses. In the retrospective patient population, 9/46 tumor samples were from patients that received chemotherapy. To determine whether chemotherapy could introduce bias in our analyses, we performed additional survival studies excluding the chemotherapy-treated patients and stratified CAV1-high versus CAV1-low (**Supplementary Fig. 7**) and statin users versus non-

statin users (**Supplementary Figure 18**). In these additional analyzes, we observed that patients with HER2⁺/CAV1^{HIGH} ($n = 11$ patients) phenotype have a worse survival than HER2⁺/CAV1^{LOW} ($n = 26$ patients); $p < 5 \times 10^{-4}$. Non-statin users ($n = 7$) have a worse survival than patients treated with statin ($n = 5$). $p=0.002$. Overall, these additional analyzes validate our findings of lower overall patient survival in CAV1-high tumors when compared with CAV1-low tumors and that statins increase survival in patients treated with Trastuzumab.

Introduction can be reduced by 50%.

Response: The reviewer makes a great point. We have modified several parts of the introduction to condense the information in the introduction section.

Patient survival (in the retrospective cohort) should also be shown simply by Her2 3+ or 2+ expression.

Response: Following the reviewer's suggestion, we performed additional patient survival analyzes stratifying patient response by HER2 IHC 3+ and IHC 2+ (**Supplementary Fig. 1**). We did not observe significant differences in patient survival in these two cohorts.

Patient population was treated with trastuzumab (and different chemo regimens) but most experiments were done with TDM-1. Complicates interpretation.

Response: Great observation and comment. Thank you.

The samples used in our studies were obtained from patients enrolled on Trastuzumab trials and 9/46 tumor samples were from patients that received other therapies prior to Trastuzumab. Patient samples used in our studies for CAV1 IHC and immunofluorescence analyzes were obtained from patients prior to their initiation of therapeutic regimens.

As mentioned in previous responses, retrospective data were collected from patients treated with Trastuzumab since we do not have data available for TDM1 in gastric cancer, since this ADC failed to treat HER2-positive gastric cancer.

Trastuzumab data is reported in our previous publication (Nature Communications, 2018), as well as in the present manuscript:

- Figure 1B shows membrane-bound and internalized Trastuzumab in HER2-expressing gastric cancer cell lines containing varying levels of CAV1;
- Figure 3G: Cell viability assays of Trastuzumab versus Trastuzumab/statin in HER2-positive gastric cancer cell lines;
- Figure 4: Trastuzumab/Statin efficacy in a humanized mouse model.

The IMPACT panel focused on Her2 only and other data are shown in a limited manner. Not clear. the number of patients with NGS is rather small. Not sure what value this analysis has provided.

Response: Considering the comments made by Reviewer #1 and Reviewer #2, we have omitted the IMPACT data comparing CAV1-high versus CAV1-low tumors, as they do not seem to add additional value to our manuscript. Kindly see our response to Comment #1 of Reviewer #1.

Majority of PDXs were pretreated with trastuzumab.

Response: None of the PDXs used in our imaging or therapeutic studies were obtained from patients pretreated with trastuzumab as we have mentioned in several parts of the manuscript.

Page 7: “Preclinical imaging studies were then performed using HER2+ gastric PDXs (78% of PDXs were obtained from patients prior initiating Trastuzumab; Table 1) ...”

Page 11: “Similar to the imaging studies reported above, therapeutic cohorts used PDXs obtained from patients prior to initiating Trastuzumab therapy.”

Page 26: “PDXs used in imaging and therapeutic experiments were obtained from patients prior to initiating Trastuzumab therapy.”

Did you check CAV1 expression heterogeneity like it occurs with Her2 expression. This can be accomplished easily by checking 2-3 patient samples from proximity. If there is considerable heterogeneity in expression (meaning 2+/3+ in one sample and negative in another) it also complicates results and makes conclusion unsteady.

Response: In our analyzes, CAV1-high cases (IHC 2+ or IHC 3+) consistently exhibited strong to moderate CAV1 membrane staining. CAV1-low cases (IHC 0 or IHC 1+) showed weak to negative CAV1 membrane staining (**Supplementary Figure 3**). At the moment, we do not have a robust sample size and optimized methods for a quantitative measure of CAV1 heterogeneity. Although CAV1 genomic and proteomic intra- and inter-tumor heterogeneity is an interesting aspect to investigate, we feel that given the extent and focus of that work it does not fall within the scope of the current manuscript and warrants separate publication at a later date.

Back to the Impact panel results, mutations are not frequent in the entire population like this. Not sure what the message is. No conclusions can be drawn from this small sample size.

You suggest that CAV1 low tumors had different signaling profile than CAV1 high tumors, however, the approach is not unbiased. unless a more comprehensive interrogation is carried out (meaning we don't know what is happening with other pathways we don't check), we cannot be certain.

Response: Similar comments were made by Reviewer #1. The IMPACT panel and western blot of pHER2, pERK, and pTyr were removed to avoid any misunderstandings.

Many results are correlative. Mechanism results are not shown.

Response: We agree with the reviewer that future studies must address the specific mechanisms by which lovastatin induces changes in caveolin-1. In our previous reports, we demonstrated that the statin effect occurs as a result of alterations in different endocytic proteins (CAV1, cavin-1, endophilin, clathrin, and dynamin; Clin Cancer Research 2020). From those studies, it is not clear if the increase in clathrin and dynamin occurs as a direct consequence of cholesterol depletion or if it occurs as a compensation mechanism induced by the depletion of caveolin-1 and cavin-1. Because caveolae are rich in cholesterol and cholesterol regulates CAV1 transcription via SRE of the CAV1 promoter, a depletion in cholesterol after statin treatment leads to CAV1 depletion (reviewed in Nat Rev Cancer 15, 225–237 (2015) <https://doi.org/10.1038/nrc3915>)).

In the present manuscript, we show that statins enhance the disruption of downstream signaling and natural killer (NK) cells-mediated ADCC. In addition to modulating cholesterol, statins have pleiotropic effects that would require a robust study of determining the specific mechanism of statins in interfering with endocytosis. In addition to a direct mechanism of statin inhibition of intratumoral HMGCoA, systemic contributions are also likely to play a role in the combined anticancer effect of statin/antibody. Therefore, further studies will explore direct and indirect mechanisms by which statins enhance trastuzumab-drug efficacy.

We have modified several sections of the manuscript. Kindly see:

Page 16: “In addition to the direct role of statins in modulating intratumoral cholesterol, future studies are necessary to determine the specific direct and indirect mechanisms of statins in enhancing antibody efficacy.”

Her2 expression (in the context of CAV1) and internalization (which has its own unique set of circumstances) are confusing results.

Response: In our previous studies, we used several different techniques and cell lines to demonstrate a role for CAV1 in mediating HER2 internalization, and that depleting CAV1 using siRNA or lovastatin decreases HER2 internalization and increase the levels of HER2 on the cell surface (Nat Commun 2018).

Changes have been made to the text to ensure clarity and avoid misunderstandings. Additionally, we removed Figure 2A of HER2/CAV1 ratio to avoid misunderstandings.

CAV1 depletion experiments are best done with trastuzumab (not TDM1). Page 9

Response: CAV1 depletion using siRNA or statins in the context of Trastuzumab were already reported in our previous studies (Nat Commun, 2018).

Results described on page 9 relate to imaging and biodistribution studies with TDM1 in CAV1-high versus CAV1-low PDXs or in the Tet-on inducible model of in vivo caveolin-1 knockdown. In our imaging and biodistribution studies, we only use 45–50 µg (2.7×10^{-10} - 3.4×10^{-10} mol) of radiolabeled TDM1. At these very low concentrations of the ADC, the amount of DM1 injected is very low and we do not expect major differences between ^{89}Zr -Trastuzumab versus ^{89}Zr -TDM1.

Correlation of CAV1 expression (should be in conjunction of Her2 expression) should be considered with trastuzumab and not TDM1. Page 9

Response: The additional experiments suggested by the reviewer are now included in our manuscript (**Kindly see Figure 1B**). The results obtained in HER2+ GC cells validate our previously reported findings in breast, bladder, and gastric cancers (Pereira 2018, Nat Commun).

Page 7: “To test Trastuzumab binding in HER2+ cancer cells expressing varying levels of CAV1, we first used a panel of GC cell lines (Fig. 1B, Supplementary Fig. 6A,B): the HER2+ GC cell line (WT, NCIN87) and three GC cell lines (AGS, KATOIII, and SNU1) stably expressing HER2 (LV-HER2). In addition to the generation of KATOIII, AGS, and SNU1 sublines stably expressing HER2, we also attempted to express HER2 in MKN45 and SNU5 GC cells. Interestingly, the protocols herein used did not allow for the successful generation of LV-HER2 in cell lines containing the highest CAV1 expression (Supplementary Fig. 6B). Membrane-bound Trastuzumab was higher in CAV1-low AGS LV-HER2 and SNU1 LV-HER2 GC cells when compared with CAV1-high NCIN87 or KATOIII LV-HER2 cells (Fig. 1B).”

Conclusions (Suppl Fig 5E) that CAV1 KO enhances TDM-1 binding. This is very confusing. Was Her2 expression increased to promote TDM-1 binding? Please clarify Page 10.

Response: In our studies, we did not perform knockout (KO) studies. A KO model would result in complete removal of CAV1, which would result in defective internalization of TDM1. Additionally, we have observed that CAV1 KO results in cell death. Instead, we developed a model of Tet-on inducible knockdown (KD) of caveolin-1. In this model, CAV1 is depleted only in the presence of doxycycline resulting in an increase in HER2 at the cell surface of cancer cells. An increase in membrane HER2 improves TDM1 binding as previously by us (Pereira P 2018, Nat Commun). Additional data on HER2 levels after CAV1 depletion is shown in **Supplementary Fig. 8B**. Kindly see:

Page 8: “Dox-induced CAV1 knockdown resulted in a 1.9-fold increase in HER2 at the cell membrane (Supplementary Fig. 8B).”

Page 8: “These results indicate that CAV1 knockdown enhances HER2 availability at the cell membrane resulting in an increase in TDM1-tumor binding in HER2+/CAV1HIGH NCIN87 xenografts.”

Additionally, Fig 3B and Suppl Fig 6A focus on internalization of TDM-1. Please provide data on trastuzumab and Her2 expression. Also provide data on CAV1 as modulated by lovastatin. Serum lipids changes would be nice and complementary.

Response: Experiments of membrane-bound and internalized trastuzumab and Her2 expression were already reported by us in Nat Commun 2018.

Data on CAV1 as modulated by lovastatin was already extensively reported by us and others (Pereira P Nat Commun 2018; Pereira P Clin Cancer Res 2020). Further to these previous publications, our present manuscript already shows CAV1 temporal modulation in tumors as mediated by lovastatin (**kindly see Fig. 3D**).

We agree with the reviewer that analyzes of changes in serum lipids would be helpful in defining the contribution of statin in depleting cholesterol. We believe that in addition to their direct, intratumoral inhibition of HMGCR, systemic contributions are also likely to contribute to anticancer effects. This work is extensive given the need for optimization of models with different cholesterol levels, the different membrane receptors, as well as the number of antibodies that we want to include in this work. Additionally, it is important to note that, unlike in humans, statin treatment does not reduce cholesterol levels in mice (kindly see doi: 10.1172/JCI19935; DOI: 10.1158/1078-0432.CCR-20-1967). Therefore, we feel that given the extent and focus of that work it does not fall within the scope of the current manuscript and warrants separate publication at a later date.

Statements around line 266 (page 11) about internalization (no change) are also confusing and need clarification.

Response: We agree with the reviewer that our previous graphs of TDM1 internalization in the presence and absence of statin were confusing. In our previous data analyzes, we normalized pHrodo signal intensity per mg of protein. We feel that the normalization should be done by considering the number of viable cells instead of the total amount of protein. Following the reviewer's comment, we performed additional TDM1 internalization studies in NCIN87, KATOIII LV-HER2, and SNU1 LV-HER2 in the presence and absence of lovastatin. Kindly, see Fig. 3B:

Page 9: "Using the pHrodo-TDM1 conjugate, we consistently observed that lovastatin increases TDM1 internalization in NCIN87, KATOIII LV-HER2, and SNU1 LV-HER2 GC cells (Fig. 3B). Additional studies with ⁸⁹Zr-labeled TDM1 demonstrate that lovastatin decreases ADC recycling to the cell membrane (Fig. 3B)."

Fig 3G results are interesting but could be dependent on multiple other variables including sample size.

Response: Figure 3G is now Figure 3F.

Figure 3F shows TDM1 binding in 6 different gastric patient derived xenografts (PDXs) in the presence and absence of lovastatin. Although we collected data for several other PDXs, we only show these 6 unique PDXs for the following four important reasons:

- 1) These PDXs were obtained from patients prior to initiating Trastuzumab therapy.
- 2) The PDXs were histologically confirmed to be carcinomas.
- 3) The PDXs had a controlled tumor growth that allowed us to perform imaging studies accounting only for specific binding and not for EPR effects that often occur due to uncontrolled tumor growth in PDXs.

To determine the sample size in each one of our cohorts, Dr. Mauguen (8th author in the manuscript), who is an Assistant Attending Biostatistician performed the following statistical considerations:

- Based on preliminary data, the average biodistribution is 9 %ID/g in the control group (sd=5), varying from 2.6% ID/g at 4h to 17.5 %ID/g at 48h. To conclude that it is 50% higher in the lovastatin group (mean = 13.5 %ID/g), with a power of 80% and a 2-sided alpha of 0.05, we need a total of 5 mice per group. With this sample size, we had a 76% power to show a significant interaction with time.

Therefore, we used 5 mice per group in our biodistribution studies.

Additionally, TDM1 tumor uptake is determined by measuring ⁸⁹Zr-TDM1 radioactivity (%ID) and normalizing by g of tumors. With this normalization, we did account for any minor differences in tumor volume between the different groups.

Line 293. It is a bit confusing. CAV1 depletion enhances TDM-1 binding (? internalization) even with incomplete Her2 expression (did Her2 expression not increase with KO/KD of CAV1. this needs to be clearly stated.

Response: We regret the oversight. The manuscript has been carefully reviewed to ensure clarity and avoid misunderstandings. We also performed additional western blot analyzes showing an increase in cell-surface HER2 after CAV1 depletion. Kindly see answers to previous comments.

Again, suggest to replace TDM-1 with tratuzumab (in Fig 3H and supply Figs. 8B/C)

Response: Figure 3H is now Figure 3G. The data requested by the reviewer is already shown in our Figure 3. As mentioned in the figure legend, please note that Trast in the yy axes refers to Trastuzumab.

Fig 4C, please provide data on CAV1 (again, trastuzumab would have been ideal)

Response: Following the reviewer's suggestion, we performed additional Western Blot analyzes of CAV1 expression in TDM1-, Statin-, or TDM1/Statin-treated tumors. Kindly see Figure 4C.

Unfortunately, we do not have tumor samples of the trastuzumab cohorts in enough quantity to perform additional western blot analyzes.

Line 336, patient with brain met had Her2 3+ and CAV-1 3+ (rather unexpected and difficult to explain). How many unique PDXs were used. Not clear.

Response: Please see our previous comment related to sample size and PDXs used in our studies. Additionally, kindly see below statistical considerations on the number of PDXs used per cohort in therapeutic studies:

- To show that the tumor size is, on average, 2.5 times as small in the TDM1/Statin group as compared to the TDM1 only group (corresponding to a 60% decrease in tumor volume) by comparing the areas under the tumor growth curves using the Vardi test, 10 mice per group were needed to reach a power of 80% with a 1-sided alpha of 0.05.

Patient survival increased if they were on statins is (similar to many metformin papers) interesting but not reliable without additional data (Her2 expression levels, etc.)

Response: As per the reviewers' comments, we performed additional patient survival analyses stratifying by HER2 IHC 2+ versus HER2 IHC 3+ and

chemotherapy-treated versus non-treated patients. The additional Kaplan Meier analyses do not change our conclusions that background statin use in patients is associated with enhanced antibody efficacy. Please kindly see our previous comments related to our ongoing studies in the context of breast, gastric, and lung tumors.

Overall, our findings, including the additional analyzes, demonstrate that statins potentiate the susceptibility of gastric cancer cells to TDM1 by modulating HER2 membrane dynamics and HER2-ADC internalization, suggesting statin as a rational therapeutic partner for anti-HER2 ADC in HER2-positive tumors.

REVIEWERS' COMMENTS

Reviewer #1 (Remarks to the Author):

With these changes, the manuscript is now acceptable for publication.

Reviewer #2 (Remarks to the Author):

I recognize the limitations of this work but since good-hearted effort is made, I am not adding more queries. Thanks

Reviewer #3 (Remarks to the Author):

A subset of gastric cancers are HER2-positive. Trastuzumab treatment is clinically beneficial, but only one third of patients respond. Previous work has found that CAV1 is important negative regulator of the membrane density of HER2 protein, that statin treatment increases HER2 membrane abundance, and that statin treatment modulates CAV2 protein.

The current manuscript finds a negative correlation between cav1 tumoral protein levels and response to TDM1 (Trastuzumab-drug conjugate). CAV1 knockdown increased the efficacy of TDM1 in preclinical models. Interestingly, cholesterol depletion is also known to reduce CAV1 levels at the membrane. Therefore, the authors also explored the utility of statins in improving the efficacy of TDM1.

Overall, the conclusions are supported by the data. There are a few concerns.

1) As pointed out, it is unlikely that circulating concentrations of statins approach those used for in vitro assays. For in vitro studies, this is fine, as statins are used as a tool. In vivo and human studies however, make it more challenging to interpret. The authors will need to better discuss these caveats.

a. Patient data, Figure 4G: alternative conclusions or caveats include (1) patients on statins have better access to screening and medical care, and therefore live longer, (2) GC patients on statins have a prolonged survival time, regardless of HER2 status or Trastuzumab treatment, or (3) the cholesterol lowering itself (ie: hepatic actions of statins) is sufficient to invoke a protective effect.

b. For in vitro studies: by inhibiting cholesterol synthesis at HMGCR, many other aspects of cellular physiology are disrupted – and it is well known that cancer cells do not fair well under these conditions. Thus, it is impossible to ascribe the effects of statins to CAV1 alone.

c. Can statin treatments be rescued by (1) overexpression of CAV1, or (2) supplementation with downstream metabolites (such as mevalonate)?. These are standard approaches when using statins

2) Implication of NK cells: The paper fails to discuss the many ways that cholesterol depletion impacts

the immune system and cancer progression. Although NK cells are implicated here (immune-compromised mice are used) – it is likely that many more mechanisms are at play in vivo. Furthermore, this is not tied back to CAV1 nor HER2.

** See Nature Portfolio's author and referees' website at www.nature.com/authors for information about policies, services and author benefits

Reviewer #1:

With these changes, the manuscript is now acceptable for publication.

Response: We are very grateful to the reviewer for the positive overview of the manuscript and constructive suggestions to improve the paper.

Reviewer #2:

I recognize the limitations of this work but since good-hearted effort is made, I am not adding more queries. Thanks

Response: We thank again the reviewer for the suggestions to improve the quality of the manuscript. We believe that, thanks to reviewer's precious inputs, the manuscript was highly improved.

Reviewer #3:

A subset of gastric cancers are HER2-positive. Trastuzumab treatment is clinically beneficial, but only one third of patients respond. Previous work has found that CAV1 is important negative regulator of the membrane density of HER2 protein, that statin treatment increases HER2 membrane abundance, and that statin treatment modulates CAV2 protein.

The current manuscript finds a negative correlation between cav1 tumoral protein levels and response to TDM1 (Trastuzumab-drug conjugate). CAV1 knockdown increased the efficacy of TDM1 in preclinical models. Interestingly, cholesterol depletion is also known to reduce CAV1 levels at the membrane. Therefore, the authors also explored the utility of statins in improving the efficacy of TDM1.

Overall, the conclusions are supported by the data. There are a few concerns.

Response: We thank the reviewer for providing additional suggestions, which have highly increased the quality of our work.

1) As pointed out, it is unlikely that circulating concentrations of statins approach those used for in vitro assays. For in vitro studies, this is fine, as statins are used as a tool. In vivo and human studies however, make it more challenging to interpret. The authors will need to better discuss these caveats.

a. Patient data, Figure 4G: alternative conclusions or caveats include (1) patients on statins have better access to screening and medical care, and therefore live longer, (2) GC patients on statins have a prolonged survival time, regardless of HER2 status or

Trastuzumab treatment, or (3) the cholesterol lowering itself (ie: hepatic actions of statins) is sufficient to invoke a protective effect.

b. For in vitro studies: by inhibiting cholesterol synthesis at HMGCR, many other aspects of cellular physiology are disrupted – and it is well known that cancer cells do not fair well under these conditions. Thus, it is impossible to ascribe the effects of statins to CAV1 alone.

Response: We thank the reviewer for highlighting and suggesting adding other limitations of our study. We have modified the discussion section according to these suggestions:

Page 15: In addition to the direct role of statins in modulating intratumoral cholesterol, future studies are necessary to determine the specific direct and indirect mechanisms of statins in enhancing antibody efficacy.

Page 16: Additionally, the patient survival analyses do not account for variables that possibly cause residual confounding observations (e.g. economic status including screening and access to medical care, physical activity, obesity, and diet). Further, the ability of statins to increase Trastuzumab efficacy may be a result of the cholesterol lowering itself (hepatic actions of statins) or may depend on the tumor's genotype, and future analyses in increased sample size are necessary to determine factors of statin-induced efficacy in combination with Trastuzumab.

Page 16: An extensive pharmacodynamic/pharmacokinetic study is required for a clinical investigation of statin use in combination with HER2-targeted therapies. Additional studies are also necessary to determine whether patients with gastric cancer on statins have a prolonged survival time, regardless of HER2 status or Trastuzumab treatment. Although previous studies have shown that statins have antitumor effects in vitro and in preclinical models, few studies have explored the relationship between statin use and the survival of patients after cancer treatments.

c. Can statin treatments be rescued by (1) overexpression of CAV1, or (2) supplementation with downstream metabolites (such as mevalonate)? These are standard approaches when using statins.

Response: Taking this very interesting and valid comment into consideration, we have included new data determining T-DM1 internalization in gastric cancer cells treated with lovastatin in the presence of mevalonate. Kindly see supplementary figure 10.

Page 8: We consistently observed that lovastatin increases TDM1 internalization in NCIN87, KATOIII LV-HER2, and SNU1 LV-HER2 GC

cells (Fig. 3B), an effect that is rescued by the addition of mevalonate to the cell culture (Supplementary Fig. 10).

As for the comment on additional studies with CAV1 overexpression, our previous data demonstrated that forced CAV1 overexpression promotes loss of HER2 at the cell membrane (doi.org/10.1038/s41467-018-07608-w).

Implication of NK cells: The paper fails to discuss the many ways that cholesterol depletion impacts the immune system and cancer progression. Although NK cells are implicated here (immune-compromised mice are used) – it is likely that many more mechanisms are at play in vivo. Furthermore, this is not tied back to CAV1 nor HER2.

Response: We agree with the reviewer that future studies are necessary to determine whether doses of statin necessary to modulate membrane HER2 result in changes in the function of NK cells. Additional sentences were added to the discussion section of our manuscript:

Pages 15/16: Other studies have demonstrated that statins also act as immunomodulatory drugs [56]. Therefore, future studies are necessary to determine whether the statin doses necessary for modulation of cell-surface HER2 induce alterations in immune cells including NK cells.